# Convergence of Adam Under Relaxed Assumptions

**Haochuan Li**
MIT
haochuan@mit.edu

**Alexander Rakhlin**
MIT
rakhlin@mit.edu

**Ali Jadbabaie**
MIT
jadbabai@mit.edu

## Abstract

In this paper, we provide a rigorous proof of convergence of the Adaptive Moment Estimate (Adam) algorithm for a wide class of optimization objectives. Despite the popularity and efficiency of the Adam algorithm in training deep neural networks, its theoretical properties are not yet fully understood, and existing convergence proofs require unrealistically strong assumptions, such as globally bounded gradients, to show the convergence to stationary points. In this paper, we show that Adam provably converges to $\epsilon$-stationary points with $\mathcal{O}(\epsilon^{-4})$ gradient complexity under far more realistic conditions. The key to our analysis is a new proof of boundedness of gradients along the optimization trajectory of Adam, under a generalized smoothness assumption according to which the local smoothness (i.e., Hessian norm when it exists) is bounded by a sub-quadratic function of the gradient norm. Moreover, we propose a variance-reduced version of Adam with an accelerated gradient complexity of $\mathcal{O}(\epsilon^{-3})$.

## 1 Introduction

In this paper, we study the non-convex unconstrained stochastic optimization problem

$$\min_x \left\{ f(x) = \mathbb{E}_\xi \left[ f(x, \xi) \right] \right\}. \tag{1}$$

The Adaptive Moment Estimation (Adam) algorithm [23] has become one of the most popular optimizers for solving (1) when $f$ is the loss for training deep neural networks. Owing to its efficiency and robustness to hyper-parameters, it is widely applied or even sometimes the default choice in many machine learning application domains such as natural language processing [44; 4; 13], generative adversarial networks [35; 21; 55], computer vision [14], and reinforcement learning [28; 33; 40]. It is also well known that Adam significantly outperforms stochastic gradient descent (SGD) for certain models like transformer [50; 24; 1].

Despite its success in practice, theoretical analyses of Adam are still limited. The original proof of convergence in [23] was later shown by [37] to contain gaps. The authors in [37] also showed that for a range of momentum parameters chosen *independently with the problem instance*, Adam does not necessarily converge even for convex objectives. However, in deep learning practice, the hyper-parameters are in fact *problem-dependent* as they are usually tuned after given the problem and weight initialization. Recently, there have been many works proving the convergence of Adam for non-convex functions with various assumptions and problem-dependent hyper-parameter choices. However, these results leave significant room for improvement. For example, [12; 19] prove the convergence to stationary points assuming the gradients are bounded by a constant, either explicitly or implicitly. On the other hand, [51; 45] consider weak assumptions, but their convergence results are still limited. See Section 2 for more detailed discussions of related works.

To address the above-mentioned gap between theory and practice, we provide a new convergence analysis of Adam *without assuming bounded gradients*, or equivalently, Lipschitzness of the objective function. In addition, we also relax the standard global smoothness assumption, i.e., the Lipschitzness of the gradient function, as it is far from being satisfied in deep neural network training. Instead, we

37th Conference on Neural Information Processing Systems (NeurIPS 2023).

consider a more general, relaxed, and non-uniform smoothness condition according to which the local smoothness (i.e., Hessian norm when it exists) around $x$ is bounded by a sub-quadratic function of the gradient norm $\|\nabla f(x)\|$ (see Definition 3.2 and Assumption 2 for the details). This generalizes the $(L_0, L_1)$ smoothness condition proposed by [49] based on language model experiments. Even though our assumptions are much weaker and more realistic, we can still obtain the same $\mathcal{O}(\epsilon^{-4})$ gradient complexity for convergence to an $\epsilon$-stationary point.

The key to our analysis is a new technique to obtain a high probability, constant upper bound on the gradients along the optimization trajectory of Adam, without assuming Lipschitzness of the objective function. In other words, it essentially turns the bounded gradient assumption into a result that can be directly proven. Bounded gradients imply bounded stepsize at each step, with which the analysis of Adam essentially reduces to the simpler analysis of AdaBound [31]. Furthermore, once the gradient boundedness is achieved, the analysis under the generalized non-uniform smoothness assumption is not much harder than that under the standard smoothness condition. We will introduce the technique in more details in Section 5. We note that the idea of bounding gradient norm along the trajectory of the optimization algorithm can be use in other problems as well. For more details, we refer the reader to our concurrent work [26] in which we present a set of new techiniques and methods for bounding gradient norm for other optimization algorithms under a generalized smoothness condition.

Another contribution of this paper is to show that the gradient complexity of Adam can be further improved with variance reduction methods. To this end, we propose a variance-reduced version of Adam by modifying its momentum update rule, inspired by the idea of the STORM algorithm [9]. Under additional generalized smoothness assumption of *the component function* $f(\cdot, \xi)$ for each $\xi$, we show that this provably accelerates the convergence with a gradient complexity of $\mathcal{O}(\epsilon^{-3})$. This rate improves upon the existing result of [47] where the authors obtain an asymptotic convergence of their approach to variance reduction for Adam in the non-convex setting, under the bounded gradient assumption.

## 1.1 Contributions

In light of the above background, we summarize our main contributions as follows.

- We develop a new analysis to show that Adam converges to stationary points under relaxed assumptions. In particular, we do not assume bounded gradients or Lipschitzness of the objective function. Furthermore, we also consider a generalized non-uniform smoothness condition where the local smoothness or Hessian norm is bounded by a *sub-quadratic* function of the gradient norm. Under these more realistic assumptions, we obtain a *dimension free* gradient complexity of $\mathcal{O}(\epsilon^{-4})$ if the gradient noise is centered and bounded.

- We generalize our analysis to the setting where the gradient noise is centered and has sub-Gaussian norm, and show the convergence of Adam with a gradient complexity of $\mathcal{O}(\epsilon^{-4} \log^{3.25}(1/\epsilon))$.

- We propose a variance-reduced version of Adam (VRAdam) with provable convergence guarantees. In particular, we obtain the accelerated $\mathcal{O}(\epsilon^{-3})$ gradient complexity.

## 2 Related work

In this section, we discuss the relevant literature related to convergence of Adam and the generalized smoothness condition, and defer additional related work on variants of Adam and variance reduction methods to Appendix A.

**Convergence of Adam.** Adam was first proposed by Kingma and Ba [23] with a theoretical convergence guarantee for convex functions. However, Reddi et al. [37] found a gap in the proof of this convergence analysis, and also constructed counter-examples for a range of hyper-parameters on which Adam does not converge. That being said, the counter-examples depend on the hyper-parameters of Adam, i.e., they are constructed after picking the hyper-parameters. Therefore, it does not rule out the possibility of obtaining convergence guarantees for problem-dependent hyper-parameters, as also pointed out by [42; 51].

Many recent works have developed convergence analyses of Adam with various assumptions and hyper-parameter choices. Zhou et al. [54] show Adam with certain hyper-parameters can work on the counter-examples of [37]. De et al. [10] prove convergence for general non-convex functions

assuming gradients are bounded and the signs of stochastic gradients are the same along the trajectory. The analysis in [12] also relies on the bounded gradient assumption. Guo et al. [19] assume the adaptive stepsize is upper and lower bounded by two constants, which is not necessarily satisfied unless assuming bounded gradients or considering the AdaBound variant [31]. [51; 45] consider very weak assumptions. However, they show either 1) "convergence" only to some neighborhood of stationary points with a constant radius, unless assuming the strong growth condition; or 2) convergence to stationary points but with a slower rate.

**Generalized smoothness condition.** Generalizing the standard smoothness condition in a variety of settings has been a focus of many recent papers. Recently, [49] proposed a generalized smoothness condition called $(L_0, L_1)$ smoothness, which assumes the local smoothness or Hessian norm is bounded by an affine function of the gradient norm. The assumption was well-validated by extensive experiments conducted on language models. Various analyses of different algorithms under this condition were later developed [48; 34; 52; 17; 38; 8]. One recent closely-related work is [45] which studies converges of Adam under the $(L_0, L_1)$ smoothness condition. However, their results are still limited, as we have mentioned above. In this paper, we consider an even more general smoothness condition where the local smoothness is bounded by a sub-quadratic function of the gradient norm, and prove the convergence of Adam under this condition. In our concurrent work [26], we further analyze various other algorithms in both convex and non-convex settings under similar generalized smoothness conditions following the same key idea of bounding gradients along the trajectory.

## 3 Preliminaries

**Notation.** Let $\|\cdot\|$ denote the Euclidean norm of a vector or spectral norm of a matrix. For any given vector $x$, we use $(x)_i$ to denote its $i$-th coordinate and $x^2$, $\sqrt{x}$, $|x|$ to denote its coordinate-wise square, square root, and absolute value respectively. For any two vectors $x$ and $y$, we use $x \odot y$ and $x/y$ to denote their coordinate-wise product and quotient respectively. We also write $x \preceq y$ or $x \succeq y$ to denote the coordinate-wise inequality between $x$ and $y$, which means $(x)_i \leq (y)_i$ or $(x)_i \geq (y)_i$ for each coordinate index $i$. For two symmetric real matrices $A$ and $B$, we say $A \preceq B$ or $A \succeq B$ if $B - A$ or $A - B$ is positive semi-definite (PSD). Given two real numbers $a, b \in \mathbb{R}$, we denote $a \wedge b := \min\{a, b\}$ for simplicity. Finally, we use $\mathcal{O}(\cdot)$, $\Theta(\cdot)$, and $\Omega(\cdot)$ for the standard big-O, big-Theta, and big-Omega notation.

### 3.1 Description of the Adam algorithm

---
**Algorithm 1** ADAM
---
1: **Input:** $\beta, \beta_{\mathrm{sq}}, \eta, \lambda, T, x_{\mathrm{init}}$
2: **Initialize** $m_0 = v_0 = 0$ and $x_1 = x_{\mathrm{init}}$
3: **for** $t = 1, \cdots, T$ **do**
4:     Draw a new sample $\xi_t$ and perform the following updates
5:     $m_t = (1 - \beta)m_{t-1} + \beta \nabla f(x_t, \xi_t)$
6:     $v_t = (1 - \beta_{\mathrm{sq}})v_{t-1} + \beta_{\mathrm{sq}}(\nabla f(x_t, \xi_t))^2$
7:     $\hat{m}_t = \frac{m_t}{1 - (1-\beta)^t}$
8:     $\hat{v}_t = \frac{v_t}{1 - (1-\beta_{\mathrm{sq}})^t}$
9:     $x_{t+1} = x_t - \frac{\eta}{\sqrt{\hat{v}_t} + \lambda} \odot \hat{m}_t$
10: **end for**
---

The formal definition of Adam proposed in [23] is shown in Algorithm 1, where Lines 5–9 describe the update rule of iterates $\{x_t\}_{1 \leq t \leq T}$. Lines 5–6 are the updates for the first and second order momentum, $m_t$ and $v_t$, respectively. In Lines 7–8, they are re-scaled to $\hat{m}_t$ and $\hat{v}_t$ in order to correct the initialization bias due to setting $m_0 = v_0 = 0$. Then the iterate is updated by $x_{t+1} = x_t - h_t \odot \hat{m}_t$ where $h_t = \eta/(\sqrt{\hat{v}_t} + \lambda)$ is the adaptive stepsize vector for some parameters $\eta$ and $\lambda$.

### 3.2 Assumptions

In what follows below, we will state our main assumptions for analysis of Adam.

#### 3.2.1 Function class

We start with a standard assumption in optimization on the objective function $f$ whose domain lies in a Euclidean space with dimension $d$.

**Assumption 1.** *The objective function $f$ is* differentiable *and* closed *within its* open domain $\mathrm{dom}(f) \subseteq \mathbb{R}^d$ *and is bounded from below, i.e., $f^* := \inf_x f(x) > -\infty$.*

*Remark* 3.1. A function $f$ is said to be closed if its sub-level set $\{x \in \mathrm{dom}(f) \mid f(x) \le a\}$ is closed for each $a \in \mathbb{R}$. In addition, a continuous function $f$ over an open domain is closed if and only $f(x)$ tends to infinity whenever $x$ approaches to the boundary of $\mathrm{dom}(f)$, which is an important condition to ensure the iterates of Adam with a small enough stepsize $\eta$ stay within the domain with high probability. Note that this condition is mild since any continuous function defined over the entire space $\mathbb{R}^d$ is closed.

Besides Assumption 1, the only additional assumption we make regarding $f$ is that its local smoothness is bounded by a sub-quadratic function of the gradient norm. More formally, we consider the following $(\rho, L_0, L_\rho)$ smoothness condition with $0 \le \rho < 2$.

**Definition 3.2.** A differentiable real-valued function $f$ is $(\rho, L_0, L_\rho)$ smooth for some constants $\rho, L_0, L_\rho \ge 0$ if the following inequality holds *almost everywhere* in $\mathrm{dom}(f)$

$$\left\| \nabla^2 f(x) \right\| \le L_0 + L_\rho \left\| \nabla f(x) \right\|^\rho .$$

*Remark* 3.3. When $\rho = 1$, Definition 3.2 reduces to the $(L_0, L_1)$ smoothness condition in [49]. When $\rho = 0$ or $L_\rho = 0$, it reduces to the standard smoothness condition.

**Assumption 2.** *The objective function $f$ is $(\rho, L_0, L_\rho)$ smooth with $0 \le \rho < 2$.*

The standard smooth function class is very restrictive as it only contains functions that are upper and lower bounded by quadratic functions. The $(L_0, L_1)$ smooth function class is more general since it also contains, e.g., univariate polynomials and exponential functions. Assumption 2 is even more general and contains univariate rational functions, double exponential functions, etc. See Appendix D.1 for the formal propositions and proofs. We also refer the reader to our concurrent work [26] for more detailed discussions of examples of $(\rho, L_0, L_\rho)$ smooth functions for different $\rho$s.

It turns out that bounded Hessian norm at a point $x$ implies local Lipschitzness of the gradient in the neighborhood around $x$. In particular, we have the following lemma.

**Lemma 3.4.** *Under Assumptions 1 and 2, for any $a > 0$ and two points $x \in \mathrm{dom}(f), y \in \mathbb{R}^d$ such that $\|y - x\| \le \frac{a}{L_0 + L_\rho(\|\nabla f(x)\| + a)^\rho}$, we have $y \in \mathrm{dom}(f)$ and*

$$\|\nabla f(y) - \nabla f(x)\| \le (L_0 + L_\rho(\|\nabla f(x)\| + a)^\rho) \cdot \|y - x\| .$$

*Remark* 3.5. Lemma 3.4 can be actually used as the definition of $(\rho, L_0, L_\rho)$ smooth functions in place of Assumption 2. Besides the local gradient Lipschitz condition, it suggests that, as long as the update at each step is small enough, the iterates will not go outside of the domain.

For the special case of $\rho = 1$, choosing $a = \max\{\|\nabla f(x)\|, L_0/L_1\}$, one can verify that the required locality size in Lemma 3.4 satisfies $\frac{a}{L_0 + L_1(\|\nabla f(x)\| + a)} \ge \frac{1}{3L_1}$. In this case, Lemma 3.4 states that $\|x - y\| \le 1/(3L_1)$ implies $\|\nabla f(y) - \nabla f(x)\| \le 2(L_0 + L_1 \|\nabla f(x)\|) \|y - x\|$. Therefore, it reduces to the local gradient Lipschitz condition for $(L_0, L_1)$ smooth functions in [49; 48] up to numerical constant factors. For $\rho \ne 1$, the proof is more involved because Grönwall's inequality used in [49; 48] no longer applies. Therefore we defer the detailed proof of Lemma 3.4 to Appendix D.2.

### 3.2.2 Stochastic gradient

We consider one of the following two assumptions on the stochastic gradient $\nabla f(x_t, \xi_t)$ in our analysis of Adam.

**Assumption 3.** *The gradient noise is centered and almost surely bounded. In particular, for some $\sigma \ge 0$ and all $t \ge 1$,*

$$\mathbb{E}_{t-1}[\nabla f(x_t, \xi_t)] = \nabla f(x_t), \quad \|\nabla f(x_t, \xi_t) - \nabla f(x_t)\| \le \sigma, \ a.s.,$$

*where $\mathbb{E}_{t-1}[\cdot] := \mathbb{E}[\cdot | \xi_1, \ldots, \xi_{t-1}]$ is the conditional expectation given $\xi_1, \ldots, \xi_{t-1}$.*

**Assumption 4.** *The gradient noise is centered with sub-Gaussian norm. In particular, for some $R \ge 0$ and all $t \ge 1$,*

$$\mathbb{E}_{t-1}[\nabla f(x_t, \xi_t)] = \nabla f(x_t), \quad \mathbb{P}_{t-1}\left( \|\nabla f(x_t, \xi_t) - \nabla f(x_t)\| \ge s \right) \le 2e^{-\frac{s^2}{2R^2}}, \ \forall s \in \mathbb{R},$$

*where $\mathbb{E}_{t-1}[\cdot] := \mathbb{E}[\cdot | \xi_1, \ldots, \xi_{t-1}]$ and $\mathbb{P}_{t-1}[\cdot] := \mathbb{P}[\cdot | \xi_1, \ldots, \xi_{t-1}]$ are the conditional expectation and probability given $\xi_1, \ldots, \xi_{t-1}$.*

Assumption 4 is strictly weaker than Assumption 3 since an almost surely bounded random variable clearly has sub-Gaussian norm, but it results in a slightly worse convergece rate up to poly-log factors (see Theorems 4.1 and 4.2). Both of them are stronger than the most standard bounded variance assumption $\mathbb{E}[\|\nabla f(x_t, \xi_t) - \nabla f(x_t)\|^2] \leq \sigma^2$ for some $\sigma \geq 0$, although Assumption 3 is actually a common assumption in existing analyses under the $(L_0, L_1)$ smoothness condition (see e.g. [49; 48]). The extension to the bounded variance assumption is challenging and a very interesting future work as it is also the assumption considered in the lower bound [3]. We suspect that such an extension would be straightforward if we consider a mini-batch version of Algorithm 1 with a batch size of $S = \Omega(\epsilon^{-2})$, since this results in a very small variance of $\mathcal{O}(\epsilon^2)$ and thus essentially reduces the analysis to the deterministic setting. However, for practical Adam with an $\mathcal{O}(1)$ batch size, the extension is challenging and we leave it as a future work.

## 4 Results

In the section, we provide our convergence results for Adam under Assumptions 1, 2, and 3 or 4. To keep the statements of the theorems concise, we first define several problem-dependent constants. First, we let $\Delta_1 := f(x_1) - f^* < \infty$ be the initial sub-optimality gap. Next, given a large enough constant $G > 0$, we define

$$r := \min\left\{\frac{1}{5L_\rho G^{\rho-1}}, \frac{1}{5(L_0^{\rho-1}L_\rho)^{1/\rho}}\right\}, \quad L := 3L_0 + 4L_\rho G^\rho, \tag{2}$$

where $L$ can be viewed as the effective smoothness constant along the trajectory if one can show $\|\nabla f(x_t)\| \leq G$ and $\|x_{t+1} - x_t\| \leq r$ at each step (see Section 5 for more detailed discussions). We will also use $c_1, c_2$ to denote some small enough numerical constants and $C_1, C_2$ to denote some large enough ones. The formal convergence results under Assumptions 1, 2, and 3 are presented in the following theorem, whose proof is deferred in Appendix E.

**Theorem 4.1.** *Suppose Assumptions 1, 2, and 3 hold. Denote $\iota := \log(1/\delta)$ for any $0 < \delta < 1$. Let $G$ be a constant satisfying $G \geq \max\left\{2\lambda, 2\sigma, \sqrt{C_1\Delta_1 L_0}, (C_1\Delta_1 L_\rho)^{\frac{1}{2-\rho}}\right\}$. Choose*

$$0 \leq \beta_{\mathrm{sq}} \leq 1, \quad \beta \leq \min\left\{1, \frac{c_1\lambda\epsilon^2}{\sigma^2 G\sqrt{\iota}}\right\}, \quad \eta \leq c_2\min\left\{\frac{r\lambda}{G}, \frac{\sigma\lambda\beta}{LG\sqrt{\iota}}, \frac{\lambda^{3/2}\beta}{L\sqrt{G}}\right\}.$$

*Let $T = \max\left\{\frac{1}{\beta^2}, \frac{C_2\Delta_1 G}{\eta\epsilon^2}\right\}$. Then with probability at least $1 - \delta$, we have $\|\nabla f(x_t)\| \leq G$ for every $1 \leq t \leq T$, and $\frac{1}{T}\sum_{t=1}^T \|\nabla f(x_t)\|^2 \leq \epsilon^2$.*

Note that $G$, the upper bound of gradients along the trajectory, is a constant that depends on $\lambda, \sigma, L_0, L_\rho$, and the initial sub-optimality gap $\Delta_1$, but not on $\epsilon$. There is no requirement on the second order momentum parameter $\beta_{\mathrm{sq}}$, although many existing works like [12; 51; 45] need certain restrictions on it. We choose very small $\beta$ and $\eta$, both of which are $\mathcal{O}(\epsilon^2)$. Therefore, from the choice of $T$, it is clear that we obtain a gradient complexity of $\mathcal{O}(\epsilon^{-4})$, where we only consider the leading term. We are not clear whether the dependence on $\epsilon$ is optimal or not, as the $\Omega(\epsilon^{-4})$ lower bound in [3] assumes the weaker bounded variance assumption than our Assumpion 3. However, it matches the state-of-the-art complexity among existing analyses of Adam.

One limitation of the dependence of our complexity on $\lambda$ is $\mathcal{O}(\lambda^{-2})$, which might be large since $\lambda$ is usually small in practice, e.g., the default choice is $\lambda = 10^{-8}$ in the PyTorch implementation. There are some existing analyses on Adam [12; 51; 45] whose rates do not depend explicitly on $\lambda$ or only depend on $\log(1/\lambda)$. However, all of them depend on $\mathrm{poly}(d)$, whereas our rate is dimension free. The dimension $d$ is also very large, especially when training transformers, for which Adam is widely used. We believe that independence on $d$ is better than that on $\lambda$, because $d$ is fixed given the architecture of the neural network but $\lambda$ is a hyper-parameter which we have the freedom to tune. In fact, based on our preliminary experimental results on CIFAR-10 shown in Figure 1, the performance of Adam is not very sensitive to the choice of $\lambda$. Although the default choice of $\lambda$ is $10^{-8}$, increasing it up to 0.01 only makes minor differences.

As discussed in Section 3.2.2, we can generalize the bounded gradient noise condition in Assumption 3 to the weaker sub-Gaussian noise condition in Assumption 4. The following theorem formally shows the convergence result under Assumptions 1, 2, and 4, whose proof is deferred in Appendix E.6.

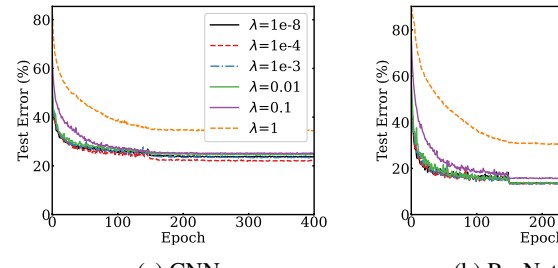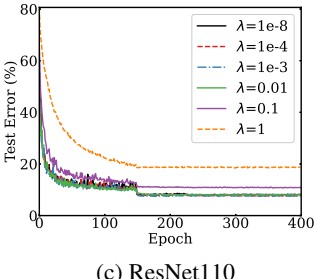

|  (a) CNN | (b) ResNet-Small | (c) ResNet110 |

Figure 1: Test errors of different models trained on CIFAR-10 using the Adam optimizer with $\beta = 0.9, \beta_{\mathrm{sq}} = 0.999, \eta = 0.001$ and different $\lambda$s. From left to right: (a) a shallow CNN with 6 layers; (b) ResNet-Small with 20 layers; and (c) ResNet110 with 110 layers.

**Theorem 4.2.** *Suppose Assumptions 1, 2, and 4 hold. Denote $\iota := \log(2/\delta)$ and $\sigma := R\sqrt{2\log(4T/\delta)}$ for any $0 < \delta < 1$. Let $G$ be a constant satisfying $G \geq \max\left\{2\lambda, 2\sigma, \sqrt{C_1\Delta_1 L_0}, (C_1\Delta_1 L_\rho)^{\frac{1}{2-\rho}}\right\}$. Choose*

$$0 \leq \beta_{\mathrm{sq}} \leq 1, \quad \beta \leq \min\left\{1, \frac{c_1\lambda\epsilon^2}{\sigma^2 G\sqrt{\iota}}\right\}, \quad \eta \leq c_2 \min\left\{\frac{r\lambda}{G}, \frac{\sigma\lambda\beta}{LG\sqrt{\iota}}, \frac{\lambda^{3/2}\beta}{L\sqrt{G}}\right\}.$$

*Let $T = \max\left\{\frac{1}{\beta^2}, \frac{C_2\Delta_1 G}{\eta\epsilon^2}\right\}$. Then with probability at least $1 - \delta$, we have $\|\nabla f(x_t)\| \leq G$ for every $1 \leq t \leq T$, and $\frac{1}{T}\sum_{t=1}^{T}\|\nabla f(x_t)\|^2 \leq \epsilon^2$.*

Note that the main difference of Theorem 4.2 from Theorem 4.1 is that $\sigma$ is now $\mathcal{O}(\sqrt{\log T})$ instead of a constant. With some standard calculations, one can show that the gradient complexity in Theorem 4.2 is bounded by $\mathcal{O}(\epsilon^{-4}\log^p(1/\epsilon))$, where $p = \max\left\{3, \frac{9+2\rho}{4}\right\} < 3.25$.

## 5 Analysis

### 5.1 Bounding the gradients along the optimization trajectory

We want to bound the gradients along the optimization trajectory mainly for two reasons. First, as discussed in Section 2, many existing analyses of Adam rely on the assumption of bounded gradients, because unbounded gradient norm leads to unbounded second order momentum $\hat{v}_t$ which implies very small stepsize, and slow convergence. On the other hand, once the gradients are bounded, it is straightforward to control $\hat{v}_t$ as well as the stepsize, and therefore the analysis essentially reduces to the easier one for AdaBound. Second, informally speaking[1], under Assumption 2, bounded gradients also imply bounded Hessians, which essentially reduces the $(\rho, L_0, L_\rho)$ smoothness to the standard smoothness. See Section 5.2 for more formal discussions.

In this paper, instead of imposing the strong assumption of globally bounded gradients, we develop a new analysis to show that with high probability, the gradients are always bounded along the trajectory of Adam until convergence. The essential idea can be informally illustrated by the following "circular" reasoning that we will make precise later. On the one hand, if $\|\nabla f(x_t)\| \leq G$ for every $t \geq 1$, it is not hard to show the gradient converges to zero based on our discussions above. On the other hand, we know that a converging sequence must be upper bounded. Therefore there exists some $G'$ such that $\|\nabla f(x_t)\| \leq G'$ for every $t \geq 1$. In other words, the bounded gradient condition implies the convergence result and the convergence result also implies the boundedness condition, forming a circular argument.

This circular argument is of course flawed. However, we can break the circularity of reasoning and rigorously prove both the bounded gradient condition and the convergence result using a contradiction

---

[1]The statement is informal because here we can only show bounded gradients and Hessians at the iterate points, which only implies local smoothness near the neighborhood of each iterate point (see Section 5.2). However, the standard smoothness condition is a stronger global condition which assumes bounded Hessian at every point within a convex set.

argument. Before introducing the contradiction argument, we first need to provide the following useful lemma, which is the reverse direction of a generalized Polyak-Lojasiewicz (PL) inequality, whose proof is deferred in Appendix D.3.

**Lemma 5.1.** *Under Assumptions 1 and 2, we have* $\|\nabla f(x)\|^2 \leq 3(3L_0 + 4L_\rho \|\nabla f(x)\|^\rho)(f(x) - f^*)$.

Define the function $\zeta(u) := \frac{u^2}{3(3L_0 + 4L_\rho u^\rho)}$ over $u \geq 0$. It is easy to verify that if $\rho < 2$, $\zeta$ is increasing and its range is $[0, \infty)$. Therefore, $\zeta$ is invertible and $\zeta^{-1}$ is also increasing. Then, for any constant $G > 0$, denoting $F = \zeta(G)$, Lemma 5.1 suggests that if $f(x) - f^* \leq F$, we have

$$\|\nabla f(x)\| \leq \zeta^{-1}(f(x) - f^*) \leq \zeta^{-1}(F) = G.$$

In other words, if $\rho < 2$, the gradient is bounded within any sub-level set, even though the sub-level set could be unbounded. Then, let $\tau$ be the first time the sub-optimality gap is strictly greater than $F$, truncated at $T + 1$, or formally,

$$\tau := \min\{t \mid f(x_t) - f^* > F\} \wedge (T + 1). \tag{3}$$

Then at least when $t < \tau$, we have $f(x_t) - f^* \leq F$ and thus $\|\nabla f(x_t)\| \leq G$. Based on our discussions above, it is not hard to analyze the updates before time $\tau$, and one can contruct some Lyapunov function to obtain an upper bound on $f(x_\tau) - f^*$. On the other hand, if $\tau \leq T$, we immediately obtain a lower bound on $f(x_\tau)$, that is $f(x_\tau) - f^* > F$, by the definition of $\tau$ in (3). If the lower bound is greater than the upper bound, it leads to a contradiction, which shows $\tau = T + 1$, i.e., the sub-optimality gap and the gradient norm are always bounded by $F$ and $G$ respectively before the algorithm terminates. We will illustrate the technique in more details in the simple deterministic setting in Section 5.3, but first, in Section 5.2, we introduce several prerequisite lemmas on the $(\rho, L_0, L_\rho)$ smoothness.

## 5.2  Local smoothness

In Section 5.1, we informally mentioned that $(\rho, L_0, L_\rho)$ smoothness essentially reduces to the standard smoothness if the gradient is bounded. In this section, we will make the statement more precise. First, note that Lemma 3.4 implies the following useful corollary.

**Corollary 5.2.** *Under Assumptions 1 and 2, for any $G > 0$ and two points $x \in \mathrm{dom}(f), y \in \mathbb{R}^d$ such that $\|\nabla f(x)\| \leq G$ and $\|y - x\| \leq r := \min\left\{\frac{1}{5L_\rho G^{\rho-1}}, \frac{1}{5(L_0^{\rho-1}L_\rho)^{1/\rho}}\right\}$, denoting $L := 3L_0 + 4L_\rho G^\rho$, we have $y \in \mathrm{dom}(f)$ and*

$$\|\nabla f(y) - \nabla f(x)\| \leq L \|y - x\|, \quad f(y) \leq f(x) + \langle \nabla f(x), y - x \rangle + \frac{L}{2} \|y - x\|^2.$$

The proof of Corollary 5.2 is deferred in Appendix D.4. Although the inequalities in Corollary 5.2 look very similar to the standard global smoothness condition with constant $L$, it is still a local condition as it requires $\|x - y\| \leq r$. Fortunately, at least before $\tau$, such a requirement is easy to satisfy for small enough $\eta$, according to the following lemma whose proof is deferred in Appendix E.5.

**Lemma 5.3.** *Under Assumption 3, if $t < \tau$ and choosing $G \geq \sigma$, we have $\|x_{t+1} - x_t\| \leq \eta D$ where $D := 2G/\lambda$.*

Then as long as $\eta \leq r/D$, we have $\|x_{t+1} - x_t\| \leq r$ which satisfies the requirement in Corollary 5.2. Then we can apply the inequalities in it in the same way as the standard smoothness condition. In other words, most classical inequalities derived for standard smooth functions also apply to $(\rho, L_0, L_\rho)$ smooth functions.

## 5.3  Warm-up: analysis in the deterministic setting

In this section, we consider the simpler deterministic setting where the stochastic gradient $\nabla f(x_t, \xi_t)$ in Algorithm 1 or (18) is replaced with the exact gradient $\nabla f(x_t)$. As discussed in Section 5.1, the key in our contradiction argument is to obtain both upper and lower bounds on $f(x_\tau) - f^*$. In the following derivations, we focus on illustrating the main idea of our analysis technique and ignore minor proof details. In addition, all of them are under Assumptions 1, 2, and 3.

In order to obtain the upper bound, we need the following two lemmas. First, denoting $\epsilon_t := \hat{m}_t - \nabla f(x_t)$, we can obtain the following informal descent lemma for deterministic Adam.

**Lemma 5.4** (Descent lemma, informal). *For any $t < \tau$, choosing $G \geq \lambda$ and a small enough $\eta$,*

$$f(x_{t+1}) - f(x_t) \lesssim -\frac{\eta}{4G} \|\nabla f(x_t)\|^2 + \frac{\eta}{2\lambda} \|\epsilon_t\|^2, \tag{4}$$

*where "$\lesssim$" omits less important terms.*

Compared with the standard descent lemma for gradient descent, there is an additional term of $\|\epsilon_t\|^2$ in Lemma 5.4. In the next lemma, we bound this term recursively.

**Lemma 5.5** (Informal). *Choosing $\beta = \Theta(\eta G^{\rho+1/2})$, if $t < \tau$, we have*

$$\|\epsilon_{t+1}\|^2 \leq (1 - \beta/4) \|\epsilon_t\|^2 + \frac{\lambda\beta}{16G} \|\nabla f(x_t)\|^2. \tag{5}$$

The proof sketches of the above two lemmas are deferred in Appendix B. Now we combine them to get the upper bound on $f(x_\tau) - f^*$. Define the function $\Phi_t := f(x_t) - f^* + \frac{2\eta}{\lambda\beta} \|\epsilon_t\|^2$. Note that for any $t < \tau$, (4)+$\frac{2\eta}{\lambda\beta} \times$(5) gives

$$\Phi_{t+1} - \Phi_t \leq -\frac{\eta}{8G} \|\nabla f(x_t)\|^2. \tag{6}$$

The above inequality shows $\Phi_t$ is non-increasing and thus a Lyapunov function. Therefore, we have

$$f(x_\tau) - f^* \leq \Phi_\tau \leq \Phi_1 = \Delta_1,$$

where in the last inequality we use $\Phi_1 = f(x_1) - f^* = \Delta_1$ since $\epsilon_1 = \hat{m}_1 - \nabla f(x_1) = 0$ in the deterministic setting.

As discussed in Section 5.1, if $\tau \leq T$, we have $F < f(x_\tau) - f^* \leq \Delta_1$. Note that we are able to choose a large enough constant $G$ so that $F = \frac{G^2}{3(3L_0 + 4L_\rho G^\rho)}$ is greater than $\Delta_1$, which leads to a contradiction and shows $\tau = T + 1$. Therefore, (6) holds for all $1 \leq t \leq T$. Taking a summation over $1 \leq t \leq T$ and re-arranging terms, we get

$$\frac{1}{T} \sum_{t=1}^{T} \|\nabla f(x_t)\|^2 \leq \frac{8G(\Phi_1 - \Phi_{T+1})}{\eta T} \leq \frac{8G\Delta_1}{\eta T} \leq \epsilon^2,$$

if choosing $T \geq \frac{8G\Delta_1}{\eta\epsilon^2}$, i.e., it shows convergence with a gradient complexity of $\mathcal{O}(\epsilon^{-2})$ since both $G$ and $\eta$ are constants independent of $\epsilon$ in the deterministic setting.

## 5.4 Extension to the stochastic setting

In this part, we briefly introduce how to extend the analysis to the more challenging stochastic setting. It becomes harder to obtain an upper bound on $f(x_\tau) - f^*$ because $\Phi_t$ is no longer non-increasing due to the existence of noise. In addition, $\tau$ defined in (3) is now a random variable. Note that all the derivations, such as Lemmas 5.4 and 5.5, are conditioned on the random event $t < \tau$. Therefore, one can not simply take a total expectation of them to show $\mathbb{E}[\Phi_t]$ is non-increasing.

Fortunately, $\tau$ is in fact a stopping time with nice properties. If the noise is almost surely bounded as in Assumption 3, by a more careful analysis, we can obtain a high probability upper bound on $f(x_\tau) - f^*$ using concentration inequalities. Then we can still obtain a contradiction and convergence under this high probability event. If the noise has sub-Gaussian norm as in Assumption 4, one can change the definition of $\tau$ to

$$\tau := \min\{t \mid f(x_t) - f^* > F\} \wedge \min\{t \mid \|\nabla f(x_t) - \nabla f(x_t, \xi_t)\| > \sigma\} \wedge (T + 1)$$

for appropriately chosen $F$ and $\sigma$. Then at least when $t < \tau$, the noise is bounded by $\sigma$. Hence we can get the same upper bound on $f(x_\tau) - f^*$ as if Assumption 3 still holds. However, when $t \leq T$, the lower bound $f(x_\tau) - f^* > F$ does not necessarily holds, which requires some more careful analyses. The details of the proofs are involved and we defer them in Appendix E.

## 6 Variance-reduced Adam

In this section, we propose a variance-reduced version of Adam (VRAdam). This new algorithm is depicted in Algorithm 2. Its main difference from the original Adam is that in the momentum update

rule (Line 6), an additional term of $(1 - \beta)(\nabla f(x_t, \xi_t) - \nabla f(x_{t-1}, \xi_t))$ is added, inspired by the STORM algorithm [9]. This term corrects the bias of $m_t$ so that it is an unbiased estimate of $\nabla f(x_t)$ in the sense of total expectation, i.e., $\mathbb{E}[m_t] = \nabla f(x_t)$. We will also show that it reduces the variance and accelerates the convergence.

Aside from the adaptive stepsize, one major difference between Algorithm 2 and STORM is that our hyper-parameters $\eta$ and $\beta$ are fixed constants whereas theirs are decreasing as a function of $t$. Choosing constant hyper-parameters requires a more accurate estimate at the initialization. That is why we use a mega-batch $\mathcal{S}_1$ to evaluate the gradient at the initial point to initialize $m_1$ and $v_1$ (Lines 2–3). In practice, one can also do a full-batch gradient evaluation at initialization. Note that there is no initialization bias for the momentum, so we do not re-scale $m_t$ and only re-scale $v_t$. We also want to point out that although the initial mega-batch gradient evaluation makes the algorithm a bit harder to implement, constant hyper-parameters are usually easier to tune and more common in training deep neural networks. It should be not hard to extend our analysis to time-decreasing $\eta$ and $\beta$ and we leave it as an interesting future work.

---

**Algorithm 2** VARIANCE-REDUCED ADAM (VRADAM)

---

1: **Input:** $\beta, \beta_{\text{sq}}, \eta, \lambda, T, S_1, x_{\text{init}}$
2: Draw a batch of samples $\mathcal{S}_1$ with size $S_1$ and use them to evaluate the gradient $\nabla f(x_{\text{init}}, \mathcal{S}_1)$.
3: **Initialize** $m_1 = \nabla f(x_{\text{init}}, \mathcal{S}_1)$, $v_1 = \beta_{\text{sq}} m_1^2$, and $x_2 = x_{\text{init}} - \frac{\eta m_1}{|m_1| + \lambda}$.
4: **for** $t = 2, \cdots, T$ **do**
5:     Draw a new sample $\xi_t$ and perform the following updates:
6:     $m_t = (1 - \beta)m_{t-1} + \beta \nabla f(x_t, \xi_t) + (1 - \beta)(\nabla f(x_t, \xi_t) - \nabla f(x_{t-1}, \xi_t))$
7:     $v_t = (1 - \beta_{\text{sq}})v_{t-1} + \beta_{\text{sq}}(\nabla f(x_t, \xi_t))^2$
8:     $\hat{v}_t = \frac{v_t}{1 - (1 - \beta_{\text{sq}})^t}$
9:     $x_{t+1} = x_t - \frac{\eta}{\sqrt{\hat{v}_t} + \lambda} \odot m_t$
10: **end for**

---

In addition to Assumption 1, we need to impose the following assumptions which can be viewed as stronger versions of Assumptions 2 and 3, respectively.

**Assumption 5.** *The objective function $f$ and the component function $f(\cdot, \xi)$ for each fixed $\xi$ are $(\rho, L_0, L_\rho)$ smooth with $0 \leq \rho < 2$.*

**Assumption 6.** *The random variables $\{\xi_t\}_{1 \leq t \leq T}$ are sampled i.i.d. from some distribution $\mathcal{P}$ such that for any $x \in \text{dom}(f)$,*

$$\mathbb{E}_{\xi \sim \mathcal{P}}[\nabla f(x, \xi)] = \nabla f(x), \quad \|\nabla f(x, \xi) - \nabla f(x)\| \leq \sigma, \; a.s.$$

*Remark* 6.1. Assumption 6 is stronger than Assumption 3. Assumption 3 applies only to the iterates generated by the algorithm, while Assumption 6 is a pointwise assumption over all $x \in \text{dom}(f)$ and further assumes an i.i.d. nature of the random variables $\{\xi_t\}_{1 \leq t \leq T}$. Also note that, similar to Adam, it is straightforward to generalize the assumption to noise with sub-Gaussian norm as in Assumption 4.

### 6.1 Analysis

In this part, we briefly discuss challenges in the analysis of VRAdam. The detailed analysis is deferred in Appendix F. Note that Corollary 5.2 requires bounded update $\|x_{t+1} - x_t\| \leq r$ at each step. For Adam, it is easy to satisfy for a small enough $\eta$ according to Lemma 5.3. However, for VRAdam, obtaining a good enough almost sure bound on the update is challenging even though the gradient noise is bounded. To bypass this difficulty, we directly impose a bound on $\|\nabla f(x_t) - m_t\|$ by changing the definition of the stopping time $\tau$, similar to how we deal with the sub-Gaussian noise condition for Adam. In particular, we define

$$\tau := \min\{t \mid \|\nabla f(x_t)\| > G\} \wedge \min\{t \mid \|\nabla f(x_t) - m_t\| > G\} \wedge (T + 1).$$

Then by definition, both $\|\nabla f(x_t)\|$ and $\|\nabla f(x_t) - m_t\|$ are bounded by $G$ before time $\tau$, which directly implies bounded update $\|x_{t+1} - x_t\|$. Of course, the new definition brings new challenges to lower bounding $f(x_\tau) - f^*$, which requires more careful analyses specific to the VRAdam algorithm. Please see Appendix F for the details.

## 6.2 Convergence guarantees for VRAdam

In the section, we provide our main results for convergence of VRAdam under Assumptions 1, 5, and 6. We consider the same definitions of problem-dependent constants $\Delta_1, r, L$ as those in Section 4 to make the statements of theorems concise. Let $c$ be a small enough numerical constant and $C$ be a large enough numerical constant. The formal convergence result is shown in the following theorem.

**Theorem 6.2.** *Suppose Assumptions 1, 5, and 6 hold. For any $0 < \delta < 1$, let $G > 0$ be a constant satisfying $G \geq \max\left\{2\lambda, 2\sigma, \sqrt{C\Delta_1 L_0/\delta}, (C\Delta_1 L_\rho/\delta)^{\frac{1}{2-\rho}}\right\}$. Choose $0 \leq \beta_{\mathrm{sq}} \leq 1$ and $\beta = a^2\eta^2$, where $a = 40L\sqrt{G}\lambda^{-3/2}$. Choose*

$$\eta \leq c \cdot \min\left\{\frac{r\lambda}{G}, \quad \frac{\lambda}{L}, \quad \frac{\lambda^2\delta}{\Delta_1 L^2}, \quad \frac{\lambda^2\sqrt{\delta}\epsilon}{\sigma G L}\right\}, \quad T = \frac{64G\Delta_1}{\eta\delta\epsilon^2}, \quad S_1 \geq \frac{1}{2\beta^2 T}.$$

*Then with probability at least $1 - \delta$, we have $\|\nabla f(x_t)\| \leq G$ for every $1 \leq t \leq T$, and $\frac{1}{T}\sum_{t=1}^{T}\|\nabla f(x_t)\|^2 \leq \epsilon^2$.*

Note that the choice of $G$, the upper bound of gradients along the trajectory of VRAdam, is very similar to that in Theorem 4.1 for Adam. The only difference is that now it also depends on the failure probability $\delta$. Similar to Theorem 4.1, there is no requirement on $\beta_{\mathrm{sq}}$ and we choose a very small $\beta = \mathcal{O}(\epsilon^2)$. However, the variance reduction technique allows us to take a larger stepsize $\eta = \mathcal{O}(\epsilon)$ (compared with $\mathcal{O}(\epsilon^2)$ for Adam) and obtain an accelerated gradient complexity of $\mathcal{O}(\epsilon^{-3})$, where we only consider the leading term. We are not sure whether it is optimal as the $\Omega(\epsilon^{-3})$ lower bound in [3] assumes the weaker bounded variance condition. However, our result significantly improves upon [47], which considers a variance-reduced version of Adam by combining Adam and SVRG [22] and only obtains asymptotic convergence in the non-convex setting. Similar to Adam, our gradient complexity for VRAdam is dimension free but its dependence on $\lambda$ is $\mathcal{O}(\lambda^{-2})$. Another limitation is that, the dependence on the failure probability $\delta$ is polynomial, worse than the poly-log dependence in Theorem 4.1 for Adam.

## 7 Conclusion and future works

In this paper, we proved the convergence of Adam and its variance-reduced version under less restrictive assumptions compared to those in the existing literature. We considered a generalized non-uniform smoothness condition, according to which the Hessian norm is bounded by a sub-quadratic function of the gradient norm almost everywhere. Instead of assuming the Lipschitzness of the objective function as in existing analyses of Adam, we use a new contradiction argument to prove that gradients are bounded by a constant along the optimization trajectory. There are several interesting future directions that one could pursue following this work.

**Relaxation of the bounded noise assumption.** Our analysis relies on the assumption of bounded noise or noise with sub-Gaussian norm. However, the existing lower bounds in [3] consider the weaker bounded variance assumption. Hence, it is not clear whether the $\mathcal{O}(\epsilon^{-4})$ complexity we obtain for Adam is tight in this setting. It will be interesting to see whether one can relax the assumption to the bounded variance setting. One may gain some insights from recent papers such as [16; 46] that analyze AdaGrad under weak noise conditions. An alternative way to show the tightness of the $\mathcal{O}(\epsilon^{-4})$ complexity is to prove a lower bound under the bounded noise assumption.

**Potential applications of our technique.** Another interesting future direction is to see if the techniques developed in this work for bounding gradients (including those in the the concurrent work [26]) can be generalized to improve the convergence results for other optimization problems and algorithms. We believe it is possible so long as the function class is well behaved and the algorithm is efficient enough so that $f(x_\tau) - f^*$ can be well bounded for some appropriately defined stopping time $\tau$.

**Understanding why Adam is better than SGD.** We want to note that our results can not explain why Adam is better than SGD for training transformers, because [26] shows that non-adaptive SGD converges with the same $\mathcal{O}(\epsilon^{-4})$ gradient complexity under even weaker conditions. It would be interesting and impactful if one can find a reasonable setting (function class, gradient oracle, etc) under which Adam or other adaptive methods provably outperform SGD.

## Acknowledgments

This work was supported, in part, by the MIT-IBM Watson AI Lab and ONR Grants N00014-20-1-2394 and N00014-23-1-2299. We also acknowledge support from DOE under grant DE-SC0022199, and NSF through awards DMS-2031883 and DMS-1953181.

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

# A  Additional related work

In this section, we discuss additional related work on variants of Adam and variance reduction methods.

**Variants of Adam.**   After Reddi et al. [37] pointed out the non-convergence issue with Adam, various variants of Adam that can be proved to converge were proposed [56; 18; 6; 5; 31; 54]. For example, AMSGrad [37] and AdaFom [6] modify the second order momentum so that it is non-decreasing. AdaBound [31] explicitly imposes upper and lower bounds on the second order momentum so that the stepsize is also bounded. AdaShift [54] uses a new estimate of the second order momentum to correct the bias. There are also some works [53; 18; 20] that provide convergence guarantees of these variants. One closely related work to ours is [47], which considers a variance-reduced version of Adam by combining Adam and SVRG [22]. However, they assume bounded gradients and can only get an asymptotic convergence in the non-convex setting.

**Variance reduction methods.**   The technique of variance reduction was introduced to accelerate convex optimization in the finite-sum setting [39; 22; 41; 32; 11]. Later, many works studied variance-reduced methods in the non-convex setting and obtained improved convergence rates for standard smooth functions. For example, SVRG and SCSG improve the $\mathcal{O}(\epsilon^{-4})$ gradient complexity of stochastic gradient descent (SGD) to $\mathcal{O}(\epsilon^{-10/3})$ [2; 36; 25]. Many new variance reduction methods [15; 43; 29; 27; 9; 30] were later proposed to further improve the complexity to $\mathcal{O}(\epsilon^{-3})$, which is optimal and matches the lower bound in [3]. Recently, [38; 7] obtained the $\mathcal{O}(\epsilon^{-3})$ complexity for the more general $(L_0, L_1)$ smooth functions. Our variance-reduced Adam is motivated by the STORM algorithm proposed by [9], where an additional term is added in the momentum update to correct the bias and reduce the variance.

# B  Proof sketches of informal lemmas in Section 5.3

In this section, we provide the proof sketches of the informal lemmas in Section 5.3. We focus on illustrating the ideas rather than rigorous proof details. Please see Appendix E for more rigorous and detailed proofs of Adam in the stochastic setting.

*Proof Sketch of Lemma 5.4.* By the definition of $\tau$, for all $t < \tau$, we have $f(x_t) - f^* \leq F$ which implies $\|\nabla f(x_t)\| \leq G$. Then from the update rule (18) in Proposition E.1 provided later in Appendix E, it is easy to verify $\hat{v}_t \preceq G^2$ since $\hat{v}_t$ is a convex combination of $\{(\nabla f(x_s))^2\}_{s \leq t}$. Let $h_t := \eta/(\sqrt{\hat{v}_t} + \lambda)$ be the stepsize vector and denote $H_t := \text{diag}(h_t)$. We know

$$\frac{\eta}{2G} I \preceq \frac{\eta}{G + \lambda} I \preceq H_t \preceq \frac{\eta}{\lambda} I. \tag{7}$$

As discussed in Section 5.2, when $\eta$ is small enough, we can apply Corollary 5.2 to obtain

$$\begin{aligned}
f(x_{t+1}) - f(x_t) &\lessgtr \langle \nabla f(x_t), x_{t+1} - x_t \rangle \\
&= -\|\nabla f(x_t)\|_{H_t}^2 - \nabla f(x_t)^\top H_t \epsilon_t \\
&\leq -\frac{1}{2} \|\nabla f(x_t)\|_{H_t}^2 + \frac{1}{2} \|\epsilon_t\|_{H_t}^2 \\
&\leq -\frac{\eta}{4G} \|\nabla f(x_t)\|^2 + \frac{\eta}{2\lambda} \|\epsilon_t\|^2,
\end{aligned}$$

where in the first (approximate) inequality we ignore the second order term $\frac{1}{2} L \|x_{t+1} - x_t\|^2 \propto \eta^2$ in Corollary 5.2 for small enough $\eta$; the equality applies the update rule $x_{t+1} - x_t = -H_t \hat{m}_t = -H_t(\nabla f(x_t) + \epsilon_t)$; in the second inequality we use $2a^\top A b \leq \|a\|_A^2 + \|b\|_A^2$ for any PSD matrix $A$ and vectors $a$ and $b$; and the last inequality is due to (7). $\qquad\square$

*Proof Sketch of Lemma 5.5.* By the update rule (18) in Proposition E.1, we have

$$\epsilon_{t+1} = (1 - \alpha_{t+1})(\epsilon_t + \nabla f(x_t) - \nabla f(x_{t+1})). \tag{8}$$

For small enough $\eta$, we can apply Corollary 5.2 to get

$$\|\nabla f(x_{t+1}) - \nabla f(x_t)\|^2 \le L^2 \|x_{t+1} - x_t\|^2 \le \mathcal{O}(\eta^2 G^{2\rho}) \|\hat{m}_t\|^2 \le \mathcal{O}(\eta^2 G^{2\rho})(\|\nabla f(x_t)\|^2 + \|\epsilon_t\|^2), \tag{9}$$

where the second inequality is due to $L = \mathcal{O}(G^\rho)$ and $\|x_{t+1} - x_t\| = \mathcal{O}(\eta) \|\hat{m}_t\|$; and the last inequality uses $\hat{m}_t = \nabla f(x_t) + \epsilon_t$ and Young's inequality $\|a + b\|^2 \le 2 \|a\|^2 + 2 \|b\|^2$. Therefore,

$$
\begin{aligned}
\|\epsilon_{t+1}\|^2 &\le (1 - \alpha_{t+1})(1 + \alpha_{t+1}/2) \|\epsilon_t\|^2 + (1 + 2/\alpha_{t+1}) \|\nabla f(x_{t+1}) - \nabla f(x_t)\|^2 \\
&\le (1 - \alpha_{t+1}/2) \|\epsilon_t\|^2 + \mathcal{O}(\eta^2 G^{2\rho}/\alpha_{t+1}) \left( \|\nabla f(x_t)\|^2 + \|\epsilon_t\|^2 \right) \\
&\le (1 - \beta/4) \|\epsilon_t\|^2 + \frac{\lambda\beta}{16G} \|\nabla f(x_t)\|^2,
\end{aligned}
$$

where the first inequality uses (8) and Young's inequality $\|a + b\|^2 \le (1 + u) \|a\|^2 + (1 + 1/u) \|b\|^2$ for any $u > 0$; the second inequality uses $(1 - \alpha_{t+1})(1 + \alpha_{t+1}/2) \le 1 - \alpha_{t+1}/2$ and (9); and in the last inequality we use $\beta \le \alpha_{t+1}$ and choose $\beta = \Theta(\eta G^{\rho+1/2})$ which implies $\mathcal{O}(\eta^2 G^{2\rho}/\alpha_{t+1}) \le \frac{\lambda\beta}{16G} \le \beta/4$. $\qquad \square$

## C  Probabilistic lemmas

In this section, we state several well-known and useful probabilistic lemmas without proof.

**Lemma C.1** (Azuma-Hoeffding inequality). *Let $\{Z_t\}_{t \ge 1}$ be a martingale with respect to a filtration $\{\mathcal{F}_t\}_{t \ge 0}$. Assume that $|Z_t - Z_{t-1}| \le c_t$ almost surely for all $t \ge 0$. Then for any fixed $T$, with probability at least $1 - \delta$,*

$$Z_T - Z_0 \le \sqrt{2 \sum_{t=1}^{T} c_t^2 \log(1/\delta)}.$$

**Lemma C.2** (Optional Stopping Theorem). *Let $\{Z_t\}_{t \ge 1}$ be a martingale with respect to a filtration $\{\mathcal{F}_t\}_{t \ge 0}$. Let $\tau$ be a bounded stopping time with respect to the same filtration. Then we have $\mathbb{E}[Z_\tau] = \mathbb{E}[Z_0]$.*

## D  Proofs related to $(\rho, L_0, L_\rho)$ smoothness

In this section, we provide proofs related to $(\rho, L_0, L_\rho)$ smoothness. In what follows, we first provide a formal proposition in Appendix D.1 showing that univariate rational functions and double exponential functions are $(\rho, L_0, L_\rho)$ smooth with $\rho < 2$, as we claimed in Section 3.2.1, and then provide the proofs of Lemma 3.4, Lemma 5.1, and Corollary 5.2 in Appendix D.2, D.3 and D.4 respectively.

### D.1  Examples

**Proposition D.1.** *Any univariate rational function $P(x)/Q(x)$, where $P, Q$ are two polynomials, and any double exponential function $a^{(b^x)}$, where $a, b > 1$, are $(\rho, L_0, L_\rho)$ smooth with $1 < \rho < 2$. However, they are not necessarily $(L_0, L_1)$ smooth.*

*Proof of Proposition D.1.* We prove the proposition in the following four parts:

**1. Univariate rational functions are $(\rho, L_0, L_\rho)$ smooth with $1 < \rho < 2$.** Let $f(x) = P(x)/Q(x)$ where $P$ and $Q$ are two polynomials. Then the partial fractional decomposition of $f(x)$ is given by

$$f(x) = w(x) + \sum_{i=1}^{m} \sum_{r=1}^{j_i} \frac{A_{ir}}{(x - a_i)^r} + \sum_{i=1}^{n} \sum_{r=1}^{k_i} \frac{B_{ir} x + C_{ir}}{(x^2 + b_i x + c_i)^r},$$

where $w(x)$ is a polynomial, $A_{ir}, B_{ir}, C_{ir}, a_i, b_i, c_i$ are all real constants satisfying $b_i^2 - 4c_i < 0$ for each $1 \le i \le n$ which implies $x^2 + b_i x + c_i > 0$ for all $x \in \mathbb{R}$. Assume $A_{ij_i} \ne 0$ without loss of

generality. Then we know $f$ has only finite singular points $\{a_i\}_{1 \leq i \leq m}$ and has continuous first and second order derivatives at all other points. To simplify notation, denote

$$p_{ir}(x) := \frac{A_{ir}}{(x - a_i)^r}, \quad q_{ir}(x) := \frac{B_{ir}x + C_{ir}}{(x^2 + b_i x + c_i)^r}.$$

Then we have $f(x) = w(x) + \sum_{i=1}^m \sum_{r=1}^{j_i} p_{ir}(x) + \sum_{i=1}^n \sum_{r=1}^{k_i} q_{ir}(x)$. For any $3/2 < \rho < 2$, we know that $\rho > \frac{r+2}{r+1}$ for any $r \geq 1$. Then we can show that

$$\lim_{x \to a_i} \frac{|f'(x)|^\rho}{|f''(x)|} = \lim_{x \to a_i} \frac{\left|p'_{ij_i}(x)\right|^\rho}{\left|p''_{ij_i}(x)\right|} = \infty, \tag{10}$$

where the first equality is because one can easily verify that the first and second order derivatives of $p_{ij_i}$ dominate those of all other terms when $x$ goes to $a_i$, and the second equality is because $\left|p'_{ij_i}(x)\right|^\rho = \mathcal{O}\left((x - a_i)^{-\rho(j_i+1)}\right)$, $\left|p''_{ij_i}(x)\right| = \mathcal{O}\left((x - a_i)^{-(j_i+2)}\right)$, and $\rho(j_i + 1) > j_i + 2$ (here we assume $j_i \geq 1$ since otherwise there is no need to prove (10) for $i$). Note that (10) implies that, for any $L_\rho > 0$, there exists $\delta_i > 0$ such that

$$|f''(x)| \leq L_\rho |f'(x)|^\rho, \quad \text{if } |x - a_i| < \delta_i. \tag{11}$$

Similarly, one can show $\lim_{x \to \infty} \frac{|f'(x)|^\rho}{|f''(x)|} = \infty$, which implies there exists $M > 0$ such that

$$|f''(x)| \leq L_\rho |f'(x)|^\rho, \quad \text{if } |x| > M. \tag{12}$$

Define

$$\mathcal{B} := \{x \in \mathbb{R} \mid |x| \leq M \text{ and } |x - a_i| \geq \delta_i, \forall i\}.$$

We know $\mathcal{B}$ is a compact set and therefore the continuous function $f''$ is bounded within $\mathcal{B}$, i.e., there exists some constant $L_0 > 0$ such that

$$|f''(x)| \leq L_0, \quad \text{if } x \in \mathcal{B}. \tag{13}$$

Combining (11), (12), and (13), we have shown

$$|f''(x)| \leq L_0 + L_\rho |f'(x)|^\rho, \quad \forall x \in \text{dom}(f),$$

which completes the proof of the first part.

**2. Rational functions are not necessarily $(L_0, L_1)$ smooth.** Consider the ration function $f(x) = 1/x$. Then we know that $f'(x) = -1/x^2$ and $f''(x) = 2/x^3$. Note that for any $0 < x \leq \min\{(L_0 + 1)^{-1/3}, (L_1 + 1)^{-1}\}$, we have

$$|f''(x)| = \frac{1}{x^3} + \frac{1}{x} \cdot |f'(x)| > L_0 + L_1 |f'(x)|,$$

which shows $f$ is not $(L_0, L_1)$ smooth for any $L_0, L_1 \geq 0$.

**3. Double exponential functions are $(\rho, L_0, L_\rho)$ smooth with $1 < \rho < 2$.** Let $f(x) = a^{(b^x)}$, where $a, b > 1$, be a double exponential function. Then we know that

$$f'(x) = \log(a) \log(b) b^x a^{(b^x)}, \quad f''(x) = \log(b)(\log(a)b^x + 1) \cdot f'(x).$$

For any $\rho > 1$, we have

$$\lim_{x \to +\infty} \frac{|f'(x)|^\rho}{|f''(x)|} = \lim_{x \to +\infty} \frac{|f'(x)|^{\rho-1}}{\log(b)(\log(a)b^x + 1)} = \lim_{y \to +\infty} \frac{(\log(a)\log(b)y)^{\rho-1} a^{(\rho-1)y}}{\log(b)(\log(a)y + 1)} = \infty,$$

where the first equality is a direct calculation; the second equality uses change of variable $y = b^x$; and the last equality is because exponential function grows faster than linear function. Then we complete the proof following a similar argument to that in Part 1.

**4. Double exponential functions are not necessarily $(L_0, L_1)$ smooth.** Consider the double exponential function $f(x) = e^{(e^x)}$. Then we have

$$f'(x) = e^x e^{(e^x)}, \quad f''(x) = (e^x + 1) \cdot f'(x).$$

For any $x \geq \max\{\log(L_0 + 1), \log(L_1 + 1)\}$, we can show that

$$|f''(x)| > (L_1 + 1)f'(x) > L_0 + L_1 |f'(x)|,$$

which shows $f$ is not $(L_0, L_1)$ smooth for any $L_0, L_1 \geq 0$. $\qquad\square$

## D.2 Proof of Lemma 3.4

Before proving Lemma 3.4, we need the following lemma that generalizes (a special case of) Grönwall's inequality.

**Lemma D.2.** *Let $u : [a, b] \to [0, \infty)$ and $\ell : [0, \infty) \to (0, \infty)$ be two continuous functions. Suppose $u'(t) \leq \ell(u(t))$ for all $t \in (a, b)$. Denote function $\phi(w) := \int \frac{1}{\ell(w)} dw$. We have for all $t \in [a, b]$,*

$$\phi(u(t)) \leq \phi(u(a)) - a + t.$$

*Proof of Lemma D.2.* First, by definition, we know that $\phi$ is increasing since $\phi' = \frac{1}{\ell} > 0$. Let function $v$ be the solution of the following differential equation

$$v'(t) = \ell(v(t)) \ \ \forall t \in (a, b), \quad v(a) = u(a). \tag{14}$$

It is straightforward to verify that the solution to (14) satisfies

$$\phi(v(t)) - t = \phi(u(a)) - a.$$

Then it suffices to show $\phi(u(t)) \leq \phi(v(t)) \ \forall t \in [a, b]$. Note that

$$(\phi(u(t)) - \phi(v(t)))' = \phi'(u(t))u'(t) - \phi'(v(t))v'(t) = \frac{u'(t)}{\ell(u(t))} - \frac{v'(t)}{\ell(v(t))} \leq 0,$$

where the inequality is because $u'(t) \leq \ell(u(t))$ by the assumption of this lemma and $v'(t) = \ell(v(t))$ by (14). Since $\phi(u(a)) - \phi(v(a)) = 0$, we know for all $t \in [a, b]$, $\phi(u(t)) \leq \phi(v(t))$. □

With Lemma D.2, one can bound the gradient norm within a small enough neighborhood of a given point as in the following lemma.

**Lemma D.3.** *Suppose $f$ is $(\rho, L_0, L_\rho)$ smooth for some $\rho, \rho, L_0, L_\rho \geq 0$. For any $a > 0$ and points $x, y \in \operatorname{dom}(f)$ satisfying $\|y - x\| \leq \frac{a}{L_0 + L_\rho(\|\nabla f(x)\| + a)^\rho}$, we have*

$$\|\nabla f(y)\| \leq \|\nabla f(x)\| + a.$$

*Proof of Lemma D.3.* Denote functions $z(t) := (1-t)x + ty$ and $u(t) := \|\nabla f(z(t))\|$ for $0 \leq t \leq 1$. Note that for any $0 \leq t \leq s \leq 1$, by triangle inequality,

$$u(s) - u(t) \leq \|\nabla f(z(s)) - \nabla f(z(t))\|.$$

We know that $u(t) = \|\nabla f(z(t))\|$ is differentiable since $f$ is second order differentiable[2]. Then we have

$$u'(t) = \lim_{s \downarrow t} \frac{u(s) - u(t)}{s - t} \leq \lim_{s \downarrow t} \frac{\|\nabla f(z(s)) - \nabla f(z(t))\|}{s - t} = \left\| \lim_{s \downarrow t} \frac{\nabla f(z(s)) - \nabla f(z(t))}{s - t} \right\|$$

$$\leq \left\| \nabla^2 f(z(t)) \right\| \|y - x\| \leq (L_0 + L_\rho u(t)^\rho) \|y - x\|.$$

Let $\phi(w) := \int_0^w \frac{1}{(L_0 + L_\rho v^\rho)\|y - x\|} dv$. By Lemma D.2, we know that

$$\phi(\|\nabla f(y)\|) = \phi(u(1)) \leq \phi(u(0)) + 1 = \phi(\|\nabla f(x)\|) + 1.$$

Denote $\psi(w) := \int_0^w \frac{1}{(L_0 + L_\rho v^\rho)} dv = \phi(w) \cdot \|y - x\|$. We have

$$\psi(\|\nabla f(y)\|) \leq \psi(\|\nabla f(x)\|) + \|y - x\|$$

$$\leq \psi(\|\nabla f(x)\|) + \frac{a}{L_0 + L_\rho(\|\nabla f(x)\| + a)^\rho}$$

$$\leq \int_0^{\|\nabla f(x)\|} \frac{1}{(L_0 + L_\rho v^\rho)} dv + \int_{\|\nabla f(x)\|}^{\|\nabla f(x)\| + a} \frac{1}{(L_0 + L_\rho v^\rho)} dv$$

$$= \psi(\|\nabla f(x)\| + a)$$

Since $\psi$ is increasing, we have $\|\nabla f(y)\| \leq \|\nabla f(x)\| + a$. □

---

[2]Here we assume $u(t) > 0$ for $0 < t < 1$. Otherwise, we can define $t_m = \sup\{0 < t < 1 \mid u(t) = 0\}$ and consider the interval $[t_m, 1]$ instead.

With Lemma D.3, we are ready to prove Lemma 3.4.

*Proof of Lemma 3.4.* Denote $z(t) := (1 - t)x + ty$ for some $y \in \mathbb{R}^d$ satisfying $\|y - x\| \leq \frac{a}{L_0 + L_\rho(\|\nabla f(x)\| + a)^\rho}$. We first show $y \in \text{dom}(f)$ by contradiction. Suppose $y \notin \text{dom}(f)$, let us define $t_b := \inf\{0 \leq t \leq 1 \mid z(t) \notin \mathcal{X}\}$ and $z_b := z(t_b)$. Then we know $z_b$ is a boundary point of $\mathcal{X}$. Since $f$ is a closed function with an open domain, we have

$$\lim_{t \uparrow t_b} f(z(t)) = \infty. \tag{15}$$

On the other hand, by the definition of $t_b$, we know $z(t) \in \mathcal{X}$ for every $0 \leq t < t_b$. Then by Lemma D.3, for all $0 \leq t < t_b$, we have $\|\nabla f(z(t))\| \leq \|\nabla f(x)\| + a$. Therefore for all $0 \leq t < t_b$

$$\begin{aligned}
f(z(t)) &\leq f(x) + \int_0^t \langle \nabla f(z(s)), y - x \rangle \, ds \\
&\leq f(x) + (\|\nabla f(x)\| + a) \cdot \|y - x\| \\
&< \infty,
\end{aligned}$$

which contradicts with (15). Therefore we have shown $y \in \text{dom}(f)$. We have

$$\begin{aligned}
\|\nabla f(y) - \nabla f(x)\| &= \left\| \int_0^1 \nabla^2 f(z(t)) \cdot (y - x) \, dt \right\| \\
&\leq \|y - x\| \cdot \int_0^1 (L_0 + L_\rho \|\nabla f(z(t))\|^\rho) \, dt \\
&\leq \|y - x\| \cdot (L_0 + L_\rho \cdot (\|\nabla f(x)\| + a)^\rho)
\end{aligned}$$

where the last inequality is due to Lemma D.3. $\qquad\square$

## D.3 Proof of Lemma 5.1

*Proof of Lemma 5.1.* Denote $G := \|\nabla f(x)\|$ and $L := 3L_0 + 4L_\rho G^\rho$. Let $y := x - \frac{1}{2L}\nabla f(x)$. Then we have

$$\|y - x\| = \frac{G}{2L} = \frac{G}{6L_0 + 8L_\rho G^\rho} \leq \min\left\{ \frac{1}{5L_\rho G^{\rho-1}}, \frac{1}{5(L_0^{\rho-1}L_\rho)^{1/\rho}} \right\} =: r,$$

where the inequality can be easily verified considering both cases of $G \leq (L_0/L_\rho)^{1/\rho}$ and $G \geq (L_0/L_\rho)^{1/\rho}$. Then based on Corollary 5.2, we have $y \in \text{dom}(f)$ and

$$f^* - f(x) \leq f(y) - f(x) \leq \left\langle \nabla f(x), y - x \right\rangle + \frac{L}{2}\|y - x\|^2 = \frac{3LG^2}{8} \leq \frac{LG^2}{3},$$

which completes the proof. $\qquad\square$

## D.4 Proof of Corollary 5.2

*Proof of Corollary 5.2.* First, Lemma 3.4 states that for any $a > 0$,

$$\|y - x\| \leq \frac{a}{L_0 + L_\rho \cdot (\|\nabla f(x)\| + a)^\rho} \implies \|\nabla f(y) - \nabla f(x)\| \leq (L_0 + L_\rho \cdot (\|\nabla f(x)\| + a)^\rho) \|y - x\|.$$

If $\|\nabla f(x)\| \leq G$, we choose $a = \max\{G, (L_0/L_\rho)^{1/\rho}\}$. Then it is straightforward to verify that

$$\frac{a}{L_0 + L_\rho \cdot (\|\nabla f(x)\| + a)^\rho} \geq \min\left\{ \frac{1}{5L_\rho G^{\rho-1}}, \frac{1}{5(L_0^{\rho-1}L_\rho)^{1/\rho}} \right\} =: r,$$
$$L_0 + L_\rho \cdot (\|\nabla f(x)\| + a)^\rho \leq 3L_0 + 4L_\rho G^\rho =: L.$$

Therefore we have shown for any $x, y$ satisfying $\|y - x\| \leq r$,

$$\|\nabla f(y) - \nabla f(x)\| \leq L\|y - x\|. \tag{16}$$

Next, let $z(t) := (1-t)x + ty$ for $0 \le t \le 1$. We know

$$
\begin{aligned}
f(y) - f(x) &= \int_0^1 \langle \nabla f(z(t)), y - x \rangle \, dt \\
&= \int_0^1 \langle \nabla f(x), y - x \rangle + \langle \nabla f(z(t)) - \nabla f(x), y - x \rangle \, dt \\
&\le \langle \nabla f(x), y - x \rangle + \int_0^1 L \, \|z(t) - x\| \, \|y - x\| \, dt \\
&= \langle \nabla f(x), y - x \rangle + L \, \|y - x\|^2 \int_0^1 t \, dt \\
&= \langle \nabla f(x), y - x \rangle + \frac{1}{2} L \, \|y - x\|^2 \,,
\end{aligned}
$$

where the inequality is due to (16). $\qquad\square$

## E  Convergence analysis of Adam

In this section, we provide detailed convergence analysis of Adam. We will focus on proving Theorem 4.1 under the bounded noise assumption (Assumption 3) in most parts of this section except Appendix E.6 where we will show how to generalize the results to noise with sub-Gaussian norm (Assumption 4) and provide the proof of Theorem 4.2.

For completeness, we repeat some important technical definitions here. First, we define

$$
\epsilon_t := \hat{m}_t - \nabla f(x_t) \tag{17}
$$

as the deviation of the re-scaled momentum from the actual gradient. Given a large enough constant $G$ defined in Theorem 4.1, denoting $F = \frac{G^2}{3(3L_0 + 4L_\rho G^\rho)}$, we formally define the stopping time $\tau$ as

$$
\tau := \min\{t \mid f(x_t) - f^* > F\} \wedge (T + 1),
$$

i.e., $\tau$ is the first time when the sub-optimality gap is strictly greater than $F$, truncated at $T+1$ to make sure it is bounded in order to apply Lemma C.2. Based on Lemma 5.1 and the discussions below it, we know that if $t < \tau$, we have both $f(x_t) - f^* \le F$ and $\|\nabla f(x_t)\| \le G$. It is clear to see that $\tau$ is a stopping time[3] with respect to $\{\xi_t\}_{t \ge 1}$ because the event $\{\tau \ge t\}$ is a function of $\{\xi_s\}_{s < t}$ and independent of $\{\xi_s\}_{s \ge t}$. Next, let

$$
h_t := \frac{\eta}{\sqrt{\hat{v}_t} + \lambda}
$$

be the stepsize vector and $H_t := \text{diag}(h_t)$ be the diagonal stepsize matrix. Then the update rule can be written as

$$
x_{t+1} = x_t - h_t \odot \hat{m}_t = x_t - H_t \hat{m}_t.
$$

Finally, as in Corollary 5.2 and Lemma 5.3, we define the following constants.

$$
\begin{aligned}
r &:= \min\left\{ \frac{1}{5 L_\rho G^{\rho-1}}, \frac{1}{5(L_0^{\rho-1} L_\rho)^{1/\rho}} \right\}, \\
L &:= 3L_0 + 4L_\rho G^\rho, \\
D &:= 2G/\lambda.
\end{aligned}
$$

### E.1  Equivalent update rule of Adam

The bias correction steps in Lines 7–8 make Algorithm 1 a bit complicated. In the following proposition, we provide an equivalent yet simpler update rule of Adam.

---

[3]Indeed, $\tau - 1$ is also a stopping time because $\nabla f(x_t)$ only depends on $\{\xi_s\}_{s < t}$, but that is unnecessary for our analysis.

**Proposition E.1.** *Denote $\alpha_t = \frac{\beta}{1-(1-\beta)^t}$ and $\alpha_t^{\mathrm{sq}} = \frac{\beta_{\mathrm{sq}}}{1-(1-\beta_{\mathrm{sq}})^t}$. Then the update rule in Algorithm 1 is equivalent to*

$$
\begin{aligned}
\hat{m}_t &= (1 - \alpha_t)\hat{m}_{t-1} + \alpha_t \nabla f(x_t, \xi_t), \\
\hat{v}_t &= (1 - \alpha_t^{\mathrm{sq}})\hat{v}_{t-1} + \alpha_t^{\mathrm{sq}}(\nabla f(x_t, \xi_t))^2, \\
x_{t+1} &= x_t - \frac{\eta}{\sqrt{\hat{v}_t} + \lambda} \odot \hat{m}_t,
\end{aligned}
\tag{18}
$$

*where initially we set $\hat{m}_1 = \nabla f(x_1, \xi_1)$ and $\hat{v}_1 = (\nabla f(x_1, \xi_1))^2$. Note that since $1 - \alpha_1 = 1 - \alpha_1^{\mathrm{sq}} = 0$, there is no need to define $\hat{m}_0$ and $\hat{v}_0$.*

*Proof of Proposition E.1.* Denote $Z_t = 1 - (1-\beta)^t$. Then we know $\alpha_t = \beta/Z_t$ and $m_t = Z_t \hat{m}_t$. By the momentum update rule in Algorithm 1, we have

$$
Z_t \hat{m}_t = (1-\beta)Z_{t-1}\hat{m}_{t-1} + \beta \nabla f(x_t, \xi_t).
$$

Note that $Z_t$ satisfies the following property

$$
(1-\beta)Z_{t-1} = 1 - \beta - (1-\beta)^t = Z_t - \beta.
$$

Then we have

$$
\begin{aligned}
\hat{m}_t &= \frac{Z_t - \beta}{Z_t} \cdot \hat{m}_{t-1} + \frac{\beta}{Z_t} \cdot \nabla f(x_t, \xi_t) \\
&= (1 - \alpha_t)\hat{m}_{t-1} + \alpha_t \nabla f(x_t, \xi_t).
\end{aligned}
$$

Next, we verify the initial condition. By Algorithm 1, since we set $m_0 = 0$, we have $m_1 = \beta \nabla f(x_1, \xi_1)$. Therefore we have $\hat{m}_1 = m_1/Z_1 = \nabla f(x_1, \xi_1)$ since $Z_1 = \beta$. Then the proof is completed by applying the same analysis on $v_t$ and $\hat{v}_t$. $\qquad\square$

## E.2   Useful lemmas for Adam

In this section, we list several useful lemmas for the convergence analysis. Their proofs are all deferred in Appendix E.5.

First note that when $t < \tau$, all the quantities in the algorithm are well bounded. In particular, we have the following lemma.

**Lemma E.2.** *If $t < \tau$, we have*

$$
\|\nabla f(x_t)\| \le G, \quad \|\nabla f(x_t, \xi_t)\| \le G + \sigma, \quad \|\hat{m}_t\| \le G + \sigma,
$$
$$
\hat{v}_t \preceq (G + \sigma)^2, \quad \frac{\eta}{G + \sigma + \lambda} \preceq h_t \preceq \frac{\eta}{\lambda}.
$$

Next, we provide a useful lemma regarding the time-dependent re-scaled momentum parameters in (18).

**Lemma E.3.** *Let $\alpha_t = \frac{\beta}{1-(1-\beta)^t}$, then for all $T \ge 2$, we have $\sum_{t=2}^T \alpha_t^2 \le 3(1 + \beta^2 T)$.*

In the next lemma, we provide an almost sure bound on $\epsilon_t$ in order to apply Azuma-Hoeffding inequality (Lemma C.1).

**Lemma E.4.** *Denote $\gamma_{t-1} = (1 - \alpha_t)(\epsilon_{t-1} + \nabla f(x_{t-1}) - \nabla f(x_t))$. Choosing $\eta \le \min\left\{\frac{r}{D}, \frac{\sigma\beta}{DL}\right\}$, if $t \le \tau$, we have $\|\epsilon_t\| \le 2\sigma$ and $\|\gamma_{t-1}\| \le 2\sigma$.*

Finally, the following lemma hides messy calculations and will be useful in the contradiction argument.

**Lemma E.5.** *Denote*

$$
\begin{aligned}
I_1 &:= \frac{8G}{\eta\lambda}\left(\Delta_1 \lambda + 8\sigma^2\left(\frac{\eta}{\beta} + \eta\beta T\right) + 20\sigma^2 \eta\sqrt{(1/\beta^2 + T)\iota}\right), \\
I_2 &:= \frac{8GF}{\eta} = \frac{8G^3}{3\eta L}.
\end{aligned}
$$

*Under the parameter choices in either Theorem 4.1 or Theorem 4.2, we have $I_1 \le I_2$ and $I_1/T \le \epsilon^2$.*

### E.3 Proof of Theorem 4.1

Before proving the main theorems, several important lemmas are needed. First, we provide a descent lemma for Adam.

**Lemma E.6.** *If $t < \tau$, choosing $G \geq \sigma + \lambda$ and $\eta \leq \min\left\{\frac{r}{D}, \frac{\lambda}{6L}\right\}$, we have*

$$f(x_{t+1}) - f(x_t) \leq -\frac{\eta}{4G} \|\nabla f(x_t)\|^2 + \frac{\eta}{\lambda} \|\epsilon_t\|^2.$$

*Proof of Lemma E.6.* By Lemma E.2, we have if $t < \tau$,

$$\frac{\eta I}{2G} \leq \frac{\eta I}{G + \sigma + \lambda} \preceq H_t \preceq \frac{\eta I}{\lambda}. \tag{19}$$

Since we choose $\eta \leq \frac{r}{D}$, by Lemma 5.3, we have $\|x_{t+1} - x_t\| \leq r$ if $t < \tau$. Then we can apply Corollary 5.2 to show that for any $t < \tau$,

$$
\begin{aligned}
f(x_{t+1}) - f(x_t) &\leq \langle \nabla f(x_t), x_{t+1} - x_t \rangle + \frac{L}{2} \|x_{t+1} - x_t\|^2 \\
&= -(\nabla f(x_t))^\top H_t \hat{m}_t + \frac{L}{2} \hat{m}_t^\top H_t^2 \hat{m}_t \\
&\leq -\|\nabla f(x_t)\|_{H_t}^2 - (\nabla f(x_t))^\top H_t \epsilon_t + \frac{\eta L}{2\lambda} \|\hat{m}_t\|_{H_t}^2 \\
&\leq -\frac{2}{3} \|\nabla f(x_t)\|_{H_t}^2 + \frac{3}{4} \|\epsilon_t\|_{H_t}^2 + \frac{\eta L}{\lambda} \left( \|\nabla f(x_t)\|_{H_t}^2 + \|\epsilon_t\|_{H_t}^2 \right) \\
&\leq -\frac{1}{2} \|\nabla f(x_t)\|_{H_t}^2 + \|\epsilon_t\|_{H_t}^2 \\
&\leq -\frac{\eta}{4G} \|\nabla f(x_t)\|^2 + \frac{\eta}{\lambda} \|\epsilon_t\|^2,
\end{aligned}
$$

where the second inequality uses (17) and (19); the third inequality is due to Young's inequality $a^\top A b \leq \frac{1}{3} \|a\|_A^2 + \frac{3}{4} \|b\|_A^2$ and $\|a + b\|_A^2 \leq 2 \|a\|_A^2 + 2 \|b\|_A$ for any PSD matrix $A$; the second last inequality uses $\eta \leq \frac{\lambda}{6L}$; and the last inequality is due to (19). $\qquad \square$

The following lemma bounds the sum of the error term $\|\epsilon_t\|^2$ before the stopping time $\tau$. Since its proof is complicated, we defer it in Appendix E.4.

**Lemma E.7.** *If $G \geq 2\sigma$ and $\eta \leq \min\left\{\frac{r}{D}, \frac{\lambda^{3/2}\beta}{6L\sqrt{G}}, \frac{\sigma\beta}{DL}\right\}$, with probability $1 - \delta$,*

$$\sum_{t=1}^{\tau-1} \|\epsilon_t\|^2 - \frac{\lambda}{8G} \|\nabla f(x_t)\|^2 \leq 8\sigma^2 \left(1/\beta + \beta T\right) + 20\sigma^2 \sqrt{(1/\beta^2 + T)\log(1/\delta)}.$$

Combining Lemma E.6 and Lemma E.7, we obtain the following useful lemma, which simultaneously bounds $f(x_t) - f^*$ and $\sum_{t=1}^{\tau-1} \|\nabla f(x_t)\|^2$.

**Lemma E.8.** *If $G \geq 2\max\{\lambda, \sigma\}$ and $\eta \leq \min\left\{\frac{r}{D}, \frac{\lambda^{3/2}\beta}{6L\sqrt{G}}, \frac{\sigma\beta}{DL}\right\}$, then with probability at least $1 - \delta$,*

$$
\begin{aligned}
&\sum_{t=1}^{\tau-1} \|\nabla f(x_t)\|^2 + \frac{8G}{\eta} (f(x_\tau) - f^*) \\
&\leq \frac{8G}{\eta\lambda} \left( \Delta_1 \lambda + 8\sigma^2 \left(\frac{\eta}{\beta} + \eta\beta T\right) + 20\sigma^2 \eta \sqrt{(1/\beta^2 + T)\log(1/\delta)} \right).
\end{aligned}
$$

*Proof of Lemma E.8.* By telescoping, Lemma E.6 implies

$$\sum_{t=1}^{\tau-1} 2 \|\nabla f(x_t)\|^2 - \frac{8G}{\lambda} \|\epsilon_t\|^2 \leq \frac{8G}{\eta} (f(x_1) - f(x_\tau)) \leq \frac{8\Delta_1 G}{\eta}. \tag{20}$$

Lemma E.7 could be written as

$$\sum_{t=1}^{\tau-1} \frac{8G}{\lambda} \|\epsilon_t\|^2 - \|\nabla f(x_t)\|^2 \leq \frac{8G}{\lambda} \left( 8\sigma^2 \left(1/\beta + \beta T\right) + 20\sigma^2 \sqrt{(1/\beta^2 + T)\log(1/\delta)} \right). \quad (21)$$

(20) + (21) gives the desired result. $\qquad\square$

With Lemma E.8, we are ready to complete the contradiction argument and the convergence analysis. Below we provide the proof of Theorem 4.1.

*Proof of Theorem 4.1.* According to Lemma E.8, there exists some event $\mathcal{E}$ with $\mathbb{P}(\mathcal{E}) \geq 1 - \delta$, such that conditioned on $\mathcal{E}$, we have

$$\frac{8G}{\eta}(f(x_\tau) - f^*) \leq \frac{8G}{\eta\lambda} \left( \Delta_1 \lambda + 8\sigma^2 \left( \frac{\eta}{\beta} + \eta\beta T \right) + 20\sigma^2 \eta \sqrt{(1/\beta^2 + T)\log(1/\delta)} \right) =: I_1. \quad (22)$$

By the definition of $\tau$, if $\tau \leq T$, we have

$$\frac{8G}{\eta}(f(x_\tau) - f^*) > \frac{8GF}{\eta} = \frac{8G^3}{3\eta L} =: I_2.$$

Based on Lemma E.5, we have $I_1 \leq I_2$, which leads to a contradiction. Therefore, we must have $\tau = T + 1$ conditioned on $\mathcal{E}$. Then, Lemma E.8 also implies that under $\mathcal{E}$,

$$\frac{1}{T} \sum_{t=1}^{T-1} \|\nabla f(x_t)\|^2 \leq \frac{I_1}{T} \leq \epsilon^2,$$

where the last inequality is due to Lemma E.5. $\qquad\square$

## E.4 Proof of Lemma E.7

In order to prove Lemma E.7, we need the following several lemmas.

**Lemma E.9.** *Denote* $\gamma_{t-1} = (1 - \alpha_t)(\epsilon_{t-1} + \nabla f(x_{t-1}) - \nabla f(x_t))$. *If* $G \geq 2\sigma$ *and* $\eta \leq \min\left\{ \frac{r}{D}, \frac{\lambda^{3/2}\beta}{6L\sqrt{G}} \right\}$, *we have for every* $2 \leq t \leq \tau$,

$$\|\epsilon_t\|^2 \leq \left(1 - \frac{\alpha_t}{2}\right) \|\epsilon_{t-1}\|^2 + \frac{\lambda\beta}{16G} \|\nabla f(x_{t-1})\|^2 + \alpha_t^2 \sigma^2 + 2\alpha_t \Big\langle \gamma_{t-1}, \nabla f(x_t, \xi_t) - \nabla f(x_t) \Big\rangle.$$

*Proof of Lemma E.9.* According to the update rule (18), we have

$$\begin{aligned}
\epsilon_t &= (1 - \alpha_t)(\epsilon_{t-1} + \nabla f(x_{t-1}) - \nabla f(x_t)) + \alpha_t(\nabla f(x_t, \xi_t) - \nabla f(x_t)) \\
&= \gamma_{t-1} + \alpha_t(\nabla f(x_t, \xi_t) - \nabla f(x_t)). \quad (23)
\end{aligned}$$

Since we choose $\eta \leq \frac{r}{D}$, by Lemma 5.3, we have $\|x_t - x_{t-1}\| \leq r$ if $t \leq \tau$. Therefore by Corollary 5.2, for any $2 \leq t \leq \tau$,

$$\|\nabla f(x_{t-1}) - \nabla f(x_t)\| \leq L \|x_t - x_{t-1}\| \leq \frac{\eta L}{\lambda} \|\hat{m}_{t-1}\| \leq \frac{\eta L}{\lambda} \left( \|\nabla f(x_{t-1})\| + \|\epsilon_{t-1}\| \right), \quad (24)$$

Therefore

$$\begin{aligned}
\|\gamma_{t-1}\|^2 &= \|(1 - \alpha_t)\epsilon_{t-1} + (1 - \alpha_t)(\nabla f(x_{t-1}) - \nabla f(x_t))\|^2 \\
&\leq (1 - \alpha_t)^2 (1 + \alpha_t) \|\epsilon_{t-1}\|^2 + (1 - \alpha_t)^2 \left(1 + \frac{1}{\alpha_t}\right) \|\nabla f(x_{t-1}) - \nabla f(x_t)\|^2 \\
&\leq (1 - \alpha_t) \|\epsilon_{t-1}\|^2 + \frac{1}{\alpha_t} \|\nabla f(x_{t-1}) - \nabla f(x_t)\|^2 \\
&\leq (1 - \alpha_t) \|\epsilon_{t-1}\|^2 + \frac{2\eta^2 L^2}{\lambda^2 \beta} \left( \|\nabla f(x_{t-1})\|^2 + \|\epsilon_{t-1}\|^2 \right) \\
&\leq \left(1 - \frac{\alpha_t}{2}\right) \|\epsilon_{t-1}\|^2 + \frac{\lambda\beta}{16G} \|\nabla f(x_{t-1})\|^2,
\end{aligned}$$

where the first inequality uses Young's inequality $\|a + b\|^2 \le (1 + u)\|a\|^2 + (1 + 1/u)\|b\|^2$ for any $u > 0$; the second inequality is due to

$$(1 - \alpha_t)^2 (1 + \alpha_t) = (1 - \alpha_t)(1 - \alpha_t^2) \le (1 - \alpha_t),$$

$$(1 - \alpha_t)^2 \left(1 + \frac{1}{\alpha_t}\right) = \frac{1}{\alpha_t}(1 - \alpha_t)^2 (1 + \alpha_t) \le \frac{1}{\alpha_t}(1 - \alpha_t) \le \frac{1}{\alpha_t};$$

the third inequality uses (24) and Young's inequality; and in the last inequality we choose $\eta \le \frac{\lambda^{3/2}\beta}{6L\sqrt{G}}$, which implies $\frac{2\eta^2 L^2}{\lambda^2 \beta} \le \frac{\lambda\beta}{16G} \le \frac{\beta}{2} \le \frac{\alpha_t}{2}$. Then by (23), we have

$$\|\epsilon_t\|^2 = \|\gamma_{t-1}\|^2 + 2\alpha_t \Big\langle \gamma_{t-1}, \nabla f(x_t, \xi_t) - \nabla f(x_t) \Big\rangle + \alpha_t^2 \|\nabla f(x_t, \xi_t) - \nabla f(x_t)\|^2$$

$$\le \left(1 - \frac{\alpha_t}{2}\right) \|\epsilon_{t-1}\|^2 + \frac{\lambda\beta}{16G} \|\nabla f(x_{t-1})\|^2 + \alpha_t^2 \sigma^2 + 2\alpha_t \Big\langle \gamma_{t-1}, \nabla f(x_t, \xi_t) - \nabla f(x_t) \Big\rangle.$$

$\square$

**Lemma E.10.** *Denote* $\gamma_{t-1} = (1 - \alpha_t)(\epsilon_{t-1} + \nabla f(x_{t-1}) - \nabla f(x_t))$. *If* $G \ge 2\sigma$ *and* $\eta \le \min\left\{\frac{r}{D}, \frac{\sigma\beta}{DL}\right\}$, *with probability* $1 - \delta$,

$$\sum_{t=2}^{\tau} \alpha_t \Big\langle \gamma_{t-1}, \nabla f(x_t, \xi_t) - \nabla f(x_t) \Big\rangle \le 5\sigma^2 \sqrt{(1 + \beta^2 T)\log(1/\delta)}.$$

*Proof of Lemma E.10.* First note that

$$\sum_{t=2}^{\tau} \alpha_t \Big\langle \gamma_{t-1}, \nabla f(x_t, \xi_t) - \nabla f(x_t) \Big\rangle = \sum_{t=2}^{T} \alpha_t \Big\langle \gamma_{t-1} 1_{\tau \ge t}, \nabla f(x_t, \xi_t) - \nabla f(x_t) \Big\rangle.$$

Since $\tau$ is a stopping time, we know that $1_{\tau \ge t}$ is a function of $\{\xi_s\}_{s < t}$. Also, by definition, we know $\gamma_{t-1}$ is a function of $\{\xi_s\}_{s < t}$. Then, denoting

$$X_t = \alpha_t \Big\langle \gamma_{t-1} 1_{\tau \ge t}, \nabla f(x_t, \xi_t) - \nabla f(x_t) \Big\rangle,$$

we know that $\mathbb{E}_{t-1}[X_t] = 0$, which implies $\{X_t\}_{t \le T}$ is a martingale difference sequence. Also, by Assumption 3 and Lemma E.4, we can show that for all $2 \le t \le T$,

$$|X_t| \le \alpha_t \sigma \|\gamma_{t-1} 1_{\tau \ge t}\| \le 2\alpha_t \sigma^2.$$

Then by the Azuma-Hoeffding inequality (Lemma C.1), we have with probability at least $1 - \delta$,

$$\left|\sum_{t=2}^{T} X_t\right| \le 2\sigma^2 \sqrt{2\sum_{t=2}^{T} \alpha_t^2 \log(1/\delta)} \le 5\sigma^2 \sqrt{(1 + \beta^2 T)\log(1/\delta)},$$

where in the last inequality we use Lemma E.3. $\square$

Then we are ready to prove Lemma E.7.

*Proof of Lemma E.7.* By Lemma E.9, we have for every $2 \le t \le \tau$,

$$\frac{\beta}{2} \|\epsilon_{t-1}\|^2 \le \frac{\alpha_t}{2} \|\epsilon_{t-1}\|^2 \le \|\epsilon_{t-1}\|^2 - \|\epsilon_t\|^2 + \frac{\lambda\beta}{16G} \|\nabla f(x_{t-1})\|^2 + \alpha_t^2 \sigma^2$$

$$+ 2\alpha_t \Big\langle \gamma_{t-1}, \nabla f(x_t, \xi_t) - \nabla f(x_t) \Big\rangle.$$

Taking a summation over $t$ from 2 to $\tau$, we have

$$\sum_{t=2}^{\tau} \frac{\beta}{2} \|\epsilon_{t-1}\|^2 - \frac{\lambda\beta}{16G} \|\nabla f(x_{t-1})\|^2 \le \|\epsilon_1\|^2 - \|\epsilon_\tau\|^2 + \sigma^2 \sum_{t=2}^{\tau} \alpha_t^2 + 10\sigma^2 \sqrt{(1 + \beta^2 T)\log(1/\delta)}$$

$$\le 4\sigma^2(1 + \beta^2 T) + 10\sigma^2 \sqrt{(1 + \beta^2 T)\log(1/\delta)},$$

where the first inequality uses Lemma E.10; and the second inequality uses Lemma E.3 and $\|\epsilon_1\|^2 = \|\nabla f(x_1, \xi_1) - \nabla f(x_1)\|^2 \le \sigma^2$. Then we complete the proof by multiplying both sides by $2/\beta$. $\square$

## E.5 Omitted proofs for Adam

In this section, we provide all the omitted proofs for Adam including those of Lemma 5.3 and all the lemmas in Appendix E.2.

*Proof of Lemma 5.3.* According to Lemma E.2, if $t < \tau$,

$$\|x_{t+1} - x_t\| \leq \frac{\eta}{\lambda} \|\hat{m}_t\| \leq \frac{\eta(G+\sigma)}{\lambda} \leq \frac{2\eta G}{\lambda}.$$

$\square$

*Proof of Lemma E.2.* By definition of $\tau$, we have $\|\nabla f(x_t)\| \leq G$ if $t < \tau$. Then Assumption 3 directly implies $\|\nabla f(x_t, \xi_t)\| \leq G + \sigma$. $\|\hat{m}_t\|$ can be bounded by a standard induction argument as follows. First note that $\|\hat{m}_1\| = \|\nabla f(x_1, \xi_1)\| \leq G + \sigma$. Supposing $\|\hat{m}_{k-1}\| \leq G + \sigma$ for some $k < \tau$, then we have

$$\|\hat{m}_k\| \leq (1 - \alpha_k) \|\hat{m}_{k-1}\| + \alpha_k \|\nabla f(x_k, \xi_k)\| \leq G + \sigma.$$

Then we can show $\hat{v}_t \preceq (G+\sigma)^2$ in a similar way noting that $(\nabla f(x_t, \xi_t))^2 \preceq \|\nabla f(x_t, \xi_t)\|^2 \leq (G+\sigma)^2$. Given the bound on $\hat{v}_t$, it is straight forward to bound the stepsize $h_t$. $\square$

*Proof of Lemma E.3.* First, when $t \geq 1/\beta$, we have $(1 - \beta)^t \leq 1/e$. Therefore,

$$\sum_{1/\beta \leq t \leq T} (1 - (1-\beta)^t)^{-2} \leq (1 - 1/e)^{-2} T \leq 3T.$$

Next, note that when $t < 1/\beta$, we have $(1 - \beta)^t \leq 1 - \frac{1}{2}\beta t$. Then we have

$$\sum_{2 \leq t < 1/\beta} (1 - (1-\beta)^t)^{-2} \leq \frac{4}{\beta^2} \sum_{t \geq 2} t^{-m} \leq \frac{3}{\beta^2}.$$

Therefore we have $\sum_{t=2}^{T} \alpha_t^2 \leq 3(1 + \beta^2 T)$. $\square$

*Proof of Lemma E.4.* We prove $\|\epsilon_t\| \leq 2\sigma$ for all $t \leq \tau$ by induction. First, note that for $t = 1$, we have

$$\|\epsilon_1\| = \|\nabla f(x_1, \xi_1) - \nabla f(x_1)\| \leq \sigma \leq 2\sigma.$$

Now suppose $\|\epsilon_{t-1}\| \leq 2\sigma$ for some $2 \leq t \leq \tau$. According to the update rule (18), we have

$$\epsilon_t = (1 - \alpha_t)(\epsilon_{t-1} + \nabla f(x_{t-1}) - \nabla f(x_t)) + \alpha_t(\nabla f(x_t, \xi_t) - \nabla f(x_t)),$$

which implies

$$\|\epsilon_t\| \leq (2 - \alpha_t)\sigma + \|\nabla f(x_{t-1}) - \nabla f(x_t)\|.$$

Since we choose $\eta \leq \frac{r}{D}$, by Lemma 5.3, we have $\|x_t - x_{t-1}\| \leq \eta D \leq r$ if $t \leq \tau$. Therefore by Corollary 5.2, we have for any $2 \leq t \leq \tau$,

$$\|\nabla f(x_t) - \nabla f(x_{t-1})\| \leq L \|x_t - x_{t-1}\| \leq \eta DL \leq \sigma \alpha_t,$$

where the last inequality uses the choice of $\eta$ and $\beta \leq \alpha_t$. Therefore we have $\|\epsilon_t\| \leq 2\sigma$ which completes the induction. Then it is straight forward to show

$$\|\gamma_{t-1}\| \leq (1 - \alpha_t)(2\sigma + \alpha_t \sigma) \leq 2\sigma.$$

$\square$

*Proof of Lemma E.5.* We first list all the related parameter choices below for convenience.

$$G \geq \max\left\{2\lambda, 2\sigma, \sqrt{C_1 \Delta_1 L_0}, (C_1 \Delta_1 L_\rho)^{\frac{1}{2-\rho}}\right\}, \quad \beta \leq \min\left\{1, \frac{c_1 \lambda \epsilon^2}{\sigma^2 G \sqrt{\iota}}\right\},$$

$$\eta \leq c_2 \min\left\{\frac{r\lambda}{G}, \frac{\sigma\lambda\beta}{LG\sqrt{\iota}}, \frac{\lambda^{3/2}\beta}{L\sqrt{G}}\right\}, \quad T = \max\left\{\frac{1}{\beta^2}, \frac{C_2 \Delta_1 G}{\eta\epsilon^2}\right\}.$$

We will show $I_1/I_2 \leq 1$ first. Note that if denoting $W = \frac{3L}{\lambda G^2}$, we have

$$I_1/I_2 = W\Delta_1\lambda + 8W\sigma^2\left(\frac{\eta}{\beta} + \eta\beta T\right) + 20W\sigma^2\sqrt{(\eta^2/\beta^2 + \eta^2 T)\iota},$$

Below are some facts that can be easily verified given the parameter choices.

(a) By the choice of $G$, we have $G^2 \geq 6\Delta_1(3L_0 + 4L_\rho G^\rho) = 6\Delta_1 L$ for large enough $C_1$, which implies $W \leq \frac{1}{2\Delta_1\lambda}$.

(b) By the choice of $T$, we have $\eta\beta T \leq \frac{\eta}{\beta} + \frac{C_2\Delta_1 G\beta}{\epsilon^2}$.

(c) By the choice of $T$, we have $\eta^2 T = \max\left\{\left(\frac{\eta}{\beta}\right)^2, \frac{C_2\eta\Delta_1 G}{\epsilon^2}\right\} \leq \left(\frac{\eta}{\beta}\right)^2 + \frac{C_2\Delta_1\sigma\beta}{\epsilon^2} \cdot \frac{\eta}{\beta} \leq \frac{3}{2}\left(\frac{\eta}{\beta}\right)^2 + \frac{1}{2}\left(\frac{C_2\Delta_1\sigma\beta}{\epsilon^2}\right)^2$.

(d) By the choice of $\eta$, we have $\eta/\beta \leq \frac{c_2\sigma\lambda}{LG\sqrt{\iota}}$, which implies $W\sigma^2\sqrt{\iota} \cdot \frac{\eta}{\beta} \leq \frac{3c_2\sigma^3}{G^3} \leq \frac{1}{200}$ for small enough $c_2$.

(e) By the choice of $\beta$ and (a), we have $\frac{W\sigma^2\Delta_1 G\sqrt{\iota}\beta}{\epsilon^2} \leq \frac{\sigma^2 G\sqrt{\iota}\beta}{2\lambda\epsilon^2} \leq \frac{1}{100C_2}$ for small enough $c_1$.

Therefore,

$$I_1/I_2 \leq \frac{1}{2} + 8W\sigma^2\left(\frac{2\eta}{\beta} + \frac{C_2\Delta_1 G\beta}{\epsilon^2}\right) + 20W\sigma^2\sqrt{\iota}\left(\sqrt{\frac{5\eta^2}{2\beta^2} + \frac{1}{2}\left(\frac{C_2\Delta_1\sigma\beta}{\epsilon^2}\right)^2}\right)$$

$$\leq \frac{1}{2} + 48W\sigma^2\sqrt{\iota} \cdot \frac{\eta}{\beta} + \frac{24C_2 W\sigma^2\Delta_1 G\sqrt{\iota}\beta}{\epsilon^2}$$

$$\leq 1,$$

where the first inequality is due to Facts (a-c); the second inequality uses $\sigma \leq G$, $\iota \geq 1$, and $\sqrt{a+b} \leq \sqrt{a} + \sqrt{b}$ for $a, b \geq 0$; and the last inequality is due to Facts (d-e).

Next, we will show $I_1/T \leq \epsilon^2$. We have

$$I_1/T = \frac{8G\Delta_1}{\eta T} + \frac{64\sigma^2 G}{\lambda\beta T} + \frac{64\sigma^2 G\beta}{\lambda} + \frac{160\sigma^2 G\sqrt{\iota}}{\lambda}\sqrt{\frac{1}{\beta^2 T^2} + \frac{1}{T}}$$

$$\leq \frac{8\epsilon^2}{C_2} + \frac{224\sigma^2 G\sqrt{\iota}}{\lambda\beta T} + \frac{64\sigma^2 G\beta}{\lambda} + \frac{160\sigma^2 G\sqrt{\iota}}{\lambda\sqrt{T}}$$

$$\leq \frac{8\epsilon^2}{C_2} + \frac{450\sigma^2 G\sqrt{\iota}\beta}{\lambda}$$

$$= \left(\frac{8}{C_2} + 450c_1\right)\epsilon^2$$

$$\leq \epsilon^2,$$

where in the first inequality we use $T \geq \frac{C_2\Delta_1 G}{\eta\epsilon^2}$ and $\sqrt{a+b} \leq \sqrt{a} + \sqrt{b}$ for $a, b \geq 0$; the second inequality uses $T \geq \frac{1}{\beta^2}$; the second equality uses the parameter choice of $\beta$; and in the last inequality we choose a large enough $C_2$ and small enough $c_1$. $\qquad\square$

### E.6 Proof of Theorem 4.2

*Proof of Theorem 4.2.* We define stopping time $\tau$ as follows

$$\tau_1 := \min\{t \mid f(x_t) - f^* > F\} \wedge (T+1),$$
$$\tau_2 := \min\{t \mid \|\nabla f(x_t) - \nabla f(x_t, \xi_t)\| > \sigma\} \wedge (T+1),$$
$$\tau := \min\{\tau_1, \tau_2\}.$$

Then it is straightforward to verify that $\tau_1, \tau_2, \tau$ are all stopping times.

Since we want to show $\mathbb{P}(\tau \leq T)$ is small, noting that $\{\tau \leq T\} = \{\tau = \tau_1 \leq T\} \cup \{\tau = \tau_2 \leq T\}$, it suffices to bound both $\mathbb{P}(\tau = \tau_1 \leq T)$ and $\mathbb{P}(\tau = \tau_2 \leq T)$.

First, we know that

$$
\mathbb{P}(\tau = \tau_2 \leq T) \leq \mathbb{P}(\tau_2 \leq T)
$$

$$
= \mathbb{P}\left(\bigcup_{1 \leq t \leq T} \|\nabla f(x_t) - \nabla f(x_t, \xi_t)\| > \sigma\right)
$$

$$
\leq \sum_{1 \leq t \leq T} \mathbb{P}\left(\|\nabla f(x_t) - \nabla f(x_t, \xi_t)\| > \sigma\right)
$$

$$
\leq \sum_{1 \leq t \leq T} \mathbb{E}\left[\mathbb{P}_{t-1}\left(\|\nabla f(x_t) - \nabla f(x_t, \xi_t)\| > \sigma\right)\right]
$$

$$
\leq \sum_{1 \leq t \leq T} \mathbb{E}\left[2e^{-\frac{\sigma^2}{2R^2}}\right]
$$

$$
= 2Te^{-\frac{\sigma^2}{2R^2}}
$$

$$
\leq \delta/2,
$$

where the fourth inequality uses Assumption 4; and the last inequality uses $\sigma = R\sqrt{2\log(4T/\delta)}$.

Next, if $\tau = \tau_1 \leq T$, by definition, we have $f(x_\tau) - f^* > F$, or equivalently,

$$
\frac{8G}{\eta}(f(x_\tau) - f^*) > \frac{8GF}{\eta} = \frac{8G^3}{3\eta L} =: I_2.
$$

On the other hand, since for any $t < \tau$, under the new definition of $\tau$, we still have

$$
f(x_t) - f^* \leq F, \quad \|f(x_t)\| \leq G, \quad \|\nabla f(x_t) - \nabla f(x_t, \xi_t)\| \leq \sigma.
$$

Then we know that Lemma E.8 still holds because all of its requirements are still satisfied, i.e., there exists some event $\mathcal{E}$ with $\mathbb{P}(\mathcal{E}) \leq \delta/2$, such that under its complement $\mathcal{E}^c$,

$$
\sum_{t=1}^{\tau-1} \|\nabla f(x_t)\|^2 + \frac{8G}{\eta}(f(x_\tau) - f^*) \leq \frac{8G}{\eta\lambda}\left(\Delta_1\lambda + 8\sigma^2\left(\frac{\eta}{\beta} + \eta\beta T\right) + 20\sigma^2\eta\sqrt{(1/\beta^2 + T)\iota}\right)
$$

$$
=: I_1.
$$

By Lemma E.5, we know $I_1 \leq I_2$, which suggests that $\mathcal{E}^c \cap \{\tau = \tau_1 \leq T\} = \emptyset$, i.e., $\{\tau = \tau_1 \leq T\} \subset \mathcal{E}$. Then we can show

$$
\mathbb{P}(\mathcal{E} \cup \{\tau \leq T\}) \leq \mathbb{P}(\mathcal{E}) + \mathbb{P}(\tau = \tau_2 \leq T) \leq \delta.
$$

Therefore,

$$
\mathbb{P}(\mathcal{E}^c \cap \{\tau = T + 1\}) \geq 1 - \mathbb{P}(\mathcal{E} \cup \{\tau \leq T\}) \geq 1 - \delta,
$$

and under the event $\mathcal{E}^c \cap \{\tau = T + 1\}$, we have $\tau = T + 1$ and

$$
\frac{1}{T}\sum_{t=1}^{t} \|\nabla f(x_t)\|^2 \leq I_1/T \leq \epsilon^2,
$$

where the last inequality is due to Lemma E.5. $\qquad\square$

# F Convergence analysis of VRAdam

In this section, we provide detailed convergence analysis of VRAdam and prove Theorem 6.2. To do that, we first provide some technical definitions[4]. Denote

$$
\epsilon_t := m_t - \nabla f(x_t)
$$

---

[4]Note that the same symbol for Adam and VRAdam may have different meanings.

as the deviation of the momentum from the actual gradient. From the update rule in Algorithm 2, we can write

$$\epsilon_t = (1 - \beta)\epsilon_{t-1} + W_t, \tag{25}$$

where we define

$$W_t := \nabla f(x_t, \xi_t) - \nabla f(x_t) - (1 - \beta)\left(\nabla f(x_{t-1}, \xi_t) - \nabla f(x_{t-1})\right).$$

Let $G$ be the constant defined in Theorem 6.2 and denote $F := \frac{G^2}{3(3L_0 + 4L_\rho G^\rho)}$. We define the following stopping times as discussed in Section 6.1.

$$\begin{aligned}
\tau_1 &:= \min\{t \mid f(x_t) - f^* > F\} \wedge (T + 1), \\
\tau_2 &:= \min\{t \mid \|\epsilon_t\| > G\} \wedge (T + 1), \\
\tau &:= \min\{\tau_1, \tau_2\}.
\end{aligned} \tag{26}$$

It is straight forward to verify that $\tau_1, \tau_2, \tau$ are all stopping times. Then if $t < \tau$, we have

$$f(x_t) - f^* \le F, \quad \|\nabla f(x_t)\| \le G, \quad \|\epsilon_t\| \le G.$$

Then we can also bound the update $\|x_{t+1} - x_t\| \le \eta D$ where $D = 2G/\lambda$ if $t < \tau$ (see Lemma F.3 for the details). Finally, we consider the same definition of $r$ and $L$ as those for Adam. Specifically,

$$r := \min\left\{\frac{1}{5L_\rho G^{\rho-1}}, \frac{1}{5(L_0^{\rho-1} L_\rho)^{1/\rho}}\right\}, \quad L := 3L_0 + 4L_\rho G^\rho. \tag{27}$$

### F.1  Useful lemmas

We first list several useful lemmas in this section without proofs. Their proofs are deferred later in Appendix F.3.

To start with, we provide a lemma on the local smoothness of each component function $f(\cdot, \xi)$ when the gradient of the objective function $f$ is bounded.

**Lemma F.1.** *For any constant $G \ge \sigma$ and two points $x \in \operatorname{dom}(f), y \in \mathbb{R}^d$ such that $\|\nabla f(x)\| \le G$ and $\|y - x\| \le r/2$, we have $y \in \operatorname{dom}(f)$ and*

$$\|\nabla f(y) - \nabla f(x)\| \le L \|y - x\|,$$
$$\|\nabla f(y, \xi) - \nabla f(x, \xi)\| \le 4L \|y - x\|, \quad \forall \xi,$$
$$f(y) \le f(x) + \langle \nabla f(x), y - x \rangle + \frac{1}{2} L \|y - x\|^2,$$

*where $r$ and $L$ are defined in* (27).

With the new definition of stopping time $\tau$ in (26), all the quantities in Algorithm 2 are well bounded before $\tau$. In particular, the following lemma holds.

**Lemma F.2.** *If $t < \tau$, we have*

$$\|\nabla f(x_t)\| \le G, \quad \|\nabla f(x_t, \xi_t)\| \le G + \sigma, \quad \|m_t\| \le 2G,$$
$$\hat{v}_t \preceq (G + \sigma)^2, \quad \frac{\eta}{G + \sigma + \lambda} \preceq h_t \preceq \frac{\eta}{\lambda}.$$

Next, we provide the following lemma which bounds the update at each step before $\tau$.

**Lemma F.3.** *if $t < \tau$, $\|x_{t+1} - x_t\| \le \eta D$ where $D = 2G/\lambda$.*

The following lemma bounds $\|W_t\|$ when $t \le \tau$.

**Lemma F.4.** *If $t \le \tau$, $G \ge 2\sigma$, and $\eta \le \frac{r}{2D}$,*

$$\|W_t\| \le \beta\sigma + \frac{5\eta L}{\lambda}\left(\|\nabla f(x_{t-1})\| + \|\epsilon_{t-1}\|\right).$$

Finally, we present some inequalities regarding the parameter choices, which will simplify the calculations later.

**Lemma F.5.** *Under the parameter choices in Theorem 6.2, we have*

$$\frac{2\Delta_1}{F} \le \frac{\delta}{4}, \quad \frac{\lambda\Delta_1\beta}{\eta G^2} \le \frac{\delta}{4}, \quad \eta\beta T \le \frac{\lambda\Delta_1}{8\sigma^2}, \quad \eta \le \frac{\lambda^{3/2}}{40L}\sqrt{\frac{\beta}{G}}.$$

## F.2 Proof of Theorem 6.2

Before proving the theorem, we will need to present several important lemmas. First, note that the descent lemma still holds for VRAdam.

**Lemma F.6.** *If $t < \tau$, choosing $G \geq \sigma + \lambda$ and $\eta \leq \min\left\{\frac{r}{2D}, \frac{\lambda}{6L}\right\}$, we have*

$$f(x_{t+1}) - f(x_t) \leq -\frac{\eta}{4G} \|\nabla f(x_t)\|^2 + \frac{\eta}{\lambda} \|\epsilon_t\|^2.$$

*Proof of Lemma F.6.* The proof is essentially the same as that of Lemma E.6. □

**Lemma F.7.** *Choose $G \geq \max\{2\sigma, 2\lambda\}$, $S_1 \geq \frac{1}{2\beta^2 T}$, and $\eta \leq \min\left\{\frac{r}{2D}, \frac{\lambda^{3/2}}{40L}\sqrt{\frac{\beta}{G}}\right\}$. We have*

$$\mathbb{E}\left[\sum_{t=1}^{\tau-1} \frac{\beta}{2}\|\epsilon_t\|^2 - \frac{\lambda\beta}{16G}\|\nabla f(x_t)\|^2\right] \leq 4\sigma^2\beta^2 T - \mathbb{E}[\|\epsilon_\tau\|^2].$$

*Proof of Lemma F.7.* By Lemma F.4, we have

$$\|W_t\|^2 \leq 2\sigma^2\beta^2 + \frac{100\eta^2 L^2}{\lambda^2}\left(\|\nabla f(x_{t-1})\|^2 + \|\epsilon_{t-1}\|^2\right)$$

$$\leq 2\sigma^2\beta^2 + \frac{\lambda\beta}{16G}\left(\|\nabla f(x_{t-1})\|^2 + \|\epsilon_{t-1}\|^2\right),$$

where in the second inequality we choose $\eta \leq \frac{\lambda^{3/2}}{40L}\sqrt{\frac{\beta}{G}}$. Therefore, noting that $\frac{\lambda\beta}{16G} \leq \beta/2$, by (25), we have

$$\|\epsilon_t\|^2 = (1-\beta)^2 \|\epsilon_{t-1}\|^2 + \|W_t\|^2 + (1-\beta)\left\langle\epsilon_{t-1}, W_t\right\rangle$$

$$\leq (1-\beta/2)\|\epsilon_{t-1}\|^2 + \frac{\lambda\beta}{16G}\|\nabla f(x_{t-1})\|^2 + 2\sigma^2\beta^2 + (1-\beta)\left\langle\epsilon_{t-1}, W_t\right\rangle.$$

Taking a summation over $2 \leq t \leq \tau$ and re-arranging the terms, we get

$$\sum_{t=1}^{\tau-1} \frac{\beta}{2}\|\epsilon_t\|^2 - \frac{\lambda\beta}{16G}\|\nabla f(x_t)\|^2 \leq \|\epsilon_1\|^2 - \|\epsilon_\tau\|^2 + 2\sigma^2\beta^2(\tau-1) + (1-\beta)\sum_{t=2}^{\tau}\left\langle\epsilon_{t-1}, W_t\right\rangle.$$

Taking expectations on both sides, noting that

$$\mathbb{E}\left[\sum_{t=2}^{\tau}\left\langle\epsilon_{t-1}, W_t\right\rangle\right] = 0$$

by the Optional Stopping Theorem (Lemma C.2), we have

$$\mathbb{E}\left[\sum_{t=1}^{\tau-1} \frac{\beta}{2}\|\epsilon_t\|^2 - \frac{\lambda\beta}{16G}\|\nabla f(x_t)\|^2\right] \leq 2\sigma^2\beta^2 T + \mathbb{E}[\|\epsilon_1\|^2] - \mathbb{E}[\|\epsilon_\tau\|^2] \leq 4\sigma^2\beta^2 T - \mathbb{E}[\|\epsilon_\tau\|^2],$$

where in the second inequality we choose $S_1 \geq \frac{1}{2\beta^2 T}$ which implies $\mathbb{E}[\|\epsilon_1\|^2] \leq \sigma^2/S_1 \leq 2\sigma^2\beta^2 T$. □

**Lemma F.8.** *Under the parameter choices in Theorem 6.2, we have*

$$\mathbb{E}\left[\sum_{t=1}^{\tau-1}\|\nabla f(x_t)\|^2\right] \leq \frac{16G\Delta_1}{\eta}, \quad \mathbb{E}[f(x_\tau) - f^*] \leq 2\Delta_1, \quad \mathbb{E}[\|\epsilon_\tau\|^2] \leq \frac{\lambda\Delta_1\beta}{\eta}.$$

*Proof of Lemma F.8.* First note that according to Lemma F.5, it is straight forward to verify that the parameter choices in Theorem 6.2 satisfy the requirements in Lemma F.6 and Lemma F.7. Then by Lemma F.6, if $t < \tau$,

$$f(x_{t+1}) - f(x_t) \leq -\frac{\eta}{4G}\|\nabla f(x_t)\|^2 + \frac{\eta}{\lambda}\|\epsilon_t\|^2.$$

Taking a summation over $1 \le t < \tau$, re-arranging terms, multiplying both sides by $\frac{8G}{\eta}$, and taking an expection, we get

$$\mathbb{E}\left[\sum_{t=1}^{\tau-1} 2\|\nabla f(x_t)\|^2 - \frac{8G}{\lambda}\|\epsilon_t\|^2\right] \le \frac{8G}{\eta}\mathbb{E}[f(x_1) - f(x_\tau)] \le \frac{8G}{\eta}\left(\Delta_1 - \mathbb{E}[f(x_\tau) - f^*]\right). \quad (28)$$

By Lemma F.7, we have

$$\mathbb{E}\left[\sum_{t=1}^{\tau-1} \frac{8G}{\lambda}\|\epsilon_t\|^2 - \|\nabla f(x_t)\|^2\right] \le \frac{64G\sigma^2\beta T}{\lambda} - \frac{16G}{\lambda\beta}\mathbb{E}[\|\epsilon_\tau\|^2] \le \frac{8G\Delta_1}{\eta} - \frac{16G}{\lambda\beta}\mathbb{E}[\|\epsilon_\tau\|^2], \quad (29)$$

where the last inequality is due to Lemma F.5. Then (28) + (29) gives

$$\mathbb{E}\left[\sum_{t=1}^{\tau-1} \|\nabla f(x_t)\|^2\right] + \frac{8G}{\eta}\mathbb{E}[f(x_\tau) - f^*] + \frac{16G}{\lambda\beta}\mathbb{E}[\|\epsilon_\tau\|^2] \le \frac{16G\Delta_1}{\eta},$$

which completes the proof. □

With all the above lemmas, we are ready to prove the theorem.

*Proof of Theorem 6.2.* First note that according to Lemma F.5, it is straight forward to verify that the parameter choices in Theorem 6.2 satisfy the requirements in all the lemmas for VRAdam.

Then, first note that if $\tau = \tau_1 \le T$, we know $f(x_\tau) - f^* > F$ by the definition of $\tau$. Therefore,

$$\mathbb{P}(\tau = \tau_1 \le T) \le \mathbb{P}(f(x_\tau) - f^* > F) \le \frac{\mathbb{E}[f(x_\tau) - f^*]}{F} \le \frac{2\Delta_1}{F} \le \frac{\delta}{4},$$

where the second inequality uses Markov's inequality; the third inequality is by Lemma F.8; and the last inequality is due to Lemma F.5.

Similarly, if $\tau_2 = \tau \le T$, we know $\|\epsilon_\tau\| > G$. We have

$$\mathbb{P}(\tau_2 = \tau \le T) \le \mathbb{P}(\|\epsilon_\tau\| > G) = \mathbb{P}(\|\epsilon_\tau\|^2 > G^2) \le \frac{\mathbb{E}[\|\epsilon_\tau\|^2]}{G^2} \le \frac{\lambda\Delta_1\beta}{\eta G^2} \le \frac{\delta}{4},$$

where the second inequality uses Markov's inequliaty; the third inequality is by Lemma F.8; and the last inequality is due to Lemma F.5. where the last inequality is due to Lemma F.5. Therefore,

$$\mathbb{P}(\tau \le T) \le \mathbb{P}(\tau_1 = \tau \le T) + \mathbb{P}(\tau_2 = \tau \le T) \le \frac{\delta}{2}.$$

Also, note that by Lemma F.8

$$\frac{16G\Delta_1}{\eta} \ge \mathbb{E}\left[\sum_{t=1}^{\tau-1}\|\nabla f(x_t)\|^2\right]$$

$$\ge \mathbb{P}(\tau = T+1)\mathbb{E}\left[\sum_{t=1}^{T}\|\nabla f(x_t)\|^2 \,\middle|\, \tau = T+1\right]$$

$$\ge \frac{1}{2}\mathbb{E}\left[\sum_{t=1}^{T}\|\nabla f(x_t)\|^2 \,\middle|\, \tau = T+1\right],$$

where the last inequality is due to $\mathbb{P}(\tau = T+1) = 1 - \mathbb{P}(\tau \le T) \ge 1 - \delta/2 \ge 1/2$. Then we can get

$$\mathbb{E}\left[\frac{1}{T}\sum_{t=1}^{T}\|\nabla f(x_t)\|^2 \,\middle|\, \tau = T+1\right] \le \frac{32G\Delta_1}{\eta T} \le \frac{\delta\epsilon^2}{2}.$$

Let $\mathcal{F} := \left\{\frac{1}{T}\sum_{t=1}^{T}\|\nabla f(x_t)\|^2 > \epsilon^2\right\}$ be the event of not converging to stationary points. By Markov's inequality, we have

$$\mathbb{P}(\mathcal{F}|\tau = T+1) \le \frac{\delta}{2}.$$

Therefore,

$$\mathbb{P}(\mathcal{F} \cup \{\tau \le T\}) \le \mathbb{P}(\tau \le T) + \mathbb{P}(\mathcal{F}|\tau = T+1) \le \delta,$$

i.e., with probability at least $1 - \delta$, we have both $\tau = T+1$ and $\frac{1}{T}\sum_{t=1}^{T}\|\nabla f(x_t)\|^2 \le \epsilon^2$. □

### F.3 Proofs of lemmas in Appendix F.1

*Proof of Lemma F.1.* This lemma is a direct corollary of Corollary 5.2. Note that by Assumption 6, we have $\|\nabla f(x, \xi)\| \leq G + \sigma \leq 2G$. Hence, when computing the locality size and smoothness constant for the component function $f(\cdot, \xi)$, we need to replace the constant $G$ in Corollary 5.2 with $2G$, that is why we get a smaller locality size of $r/2$ and a larger smoothness constant of $4L$. $\quad\square$

*Proof of Lemma F.2.* The bound on $\|m_t\|$ is by the definition of $\tau$ in (26). All other quantities for VRAdam are defined in the same way as those in Adam (Algorithm 1), so they have the same upper bounds as in Lemma E.2. $\quad\square$

*Proof of Lemma F.3.*

$$\|x_{t+1} - x_t\| \leq \eta \|m_t\| / \lambda \leq 2\eta G/\lambda = \eta D,$$

where the first inequality uses the update rule in Algorithm 2 and $h_t \preceq \eta/\lambda$ by Lemma F.2; the second inequality is again due to Lemma F.2. $\quad\square$

*Proof of Lemma F.4.* By the definition of $W_t$, it is easy to verify that

$$W_t = \beta(\nabla f(x_t, \xi_t) - \nabla f(x_t)) + (1 - \beta)\delta_t,$$

where

$$\delta_t = \nabla f(x_t, \xi_t) - \nabla f(x_{t-1}, \xi_t) - \nabla f(x_t) + \nabla f(x_{t-1}).$$

Then we can bound

$$
\begin{aligned}
\|\delta_t\| &\leq \|\nabla f(x_t, \xi_t) - \nabla f(x_{t-1}, \xi_t)\| + \|\nabla f(x_t) - \nabla f(x_{t-1})\| \\
&\leq 5L \|x_t - x_{t-1}\| \\
&\leq \frac{5\eta L}{\lambda} \left( \|\nabla f(x_{t-1})\| + \|\epsilon_{t-1}\| \right),
\end{aligned}
$$

where the second inequality uses Lemma F.1; and the last inequality is due to $\|x_t - x_{t-1}\| \leq \eta \|m_{t-1}\| / \lambda \leq \eta \left( \|\nabla f(x_{t-1})\| + \|\epsilon_{t-1}\| \right) / \lambda$. Then, we have

$$\|W_t\| \leq \beta\sigma + \frac{5\eta L}{\lambda} \left( \|\nabla f(x_{t-1})\| + \|\epsilon_{t-1}\| \right).$$

$\quad\square$

*Proof of Lemma F.5.* These inequalities can be obtained by direct calculations. $\quad\square$

