# OpenReview forum: "Convergence of Adam Under Relaxed Assumptions"
_NeurIPS.cc/2023/Conference — NeurIPS 2023 spotlight_

### Official Review · Reviewer_LKzM · 2023-06-22

**Soundness:** 3 good
**Presentation:** 3 good
**Contribution:** 3 good
**Rating:** 7
**Confidence:** 5

**Summary:**

The paper removes the Lipschit gradient assumption for the adaptive SGD (ASGD), making the ASGD broader to wide applications. Under the weak assumption, the authors still proved the optimal convergence rate. Moreover, they propose a variance-reduced version  with an accelerated complexity. The results are interesting and novel.

**Strengths:**

1. They further relaxed the ($L_0,L_1$)  Lipschit gradient assumption, which is more relastic.

2. They  develop a new analysis to show that ASGD can be convergent under the weak Lipschit gradient assumption but still with the same rate as existing works on ASGD. The results and techniques are novel.

3. A variance-reduced version of  ASGD is proposed with provable acceleration.

4. The authors show that the rates of ASGD and its variance-reduced version is dimension-free under certain cases.

**Weaknesses:**

1. The proof of variance-reduced version is interesting but without any numerical demonstrations. If the authors proposed an accelerated algorithm, it is convening to present some numerics because the theory is not the focus now.

2. In the contribution part, the authors claim that they do not assume bounded gradients. This is because you assume the a.s. bounded noise. which has been studied by previous work [Li and Orabona, 2019].



**Questions:**

 There are very recent works on using bounded variance noises [Matthew Faw et al. COLT 2022, Bohang Wang et al. COLT 2023]. Can the results in this paper be extended to that case?

**Limitations:**

YES

---

> ### Author Rebuttal · Authors · 2023-08-09
>
> We thank the reviewer for the positive feedback! Below we will try to address the concerns and questions.
>
> - First, we thank the reviewer for the suggestion about numerical demonstrations for VRAdam. We will consider empirically comparing the performances of Adam and the variance-reduced version of it in the revision.
> - The assumption of bounded gradient is actually independent of that of bounded noise. Even in the deterministic setting with zero noise, previous analyses of Adam cited in our paper still require strong assumptions like bounded gradients. For the paper [Li and Orabona, 2019], they are studying AdaGrad which is different from Adam, and therefore not directly comparable to our results.
> - The recent works  [Matthew Faw et al. COLT 2022, Bohang Wang et al. COLT 2023] also study AdaGrad (or AdaGrad-Norm) instead of Adam. Based on our understanding, they actually rely on some properties of AdaGrad to relax the noise assumption to bounded variance. Our approach is general and can also be applied to get a convergence result for AdaGrad. However, it may not be as good as their analyses designed for AdaGrad (e.g., we may still require the bounded noise assumption for AdaGrad).
>
> Note that the reviewer did not provide enough details for us to precisely identify the references in the comments. We assume they refer to the following three papers. Please let us know if not. We will also cite them in the revision.
>
> [1] Li and Orabona. "On the Convergence of Stochastic Gradient Descent with Adaptive Stepsizes." AISTATS 2019
>
> [2] Faw et al. COLT 2022. "The Power of Adaptivity in SGD: Self-Tuning Step Sizes with Unbounded Gradients and Affine Variance." COLT 2022
>
> [3] Wang et al. "Convergence of AdaGrad for Non-convex Objectives: Simple Proofs and Relaxed Assumptions." COLT 2023

---

### Official Review · Reviewer_rhLM · 2023-07-05

**Soundness:** 4 excellent
**Presentation:** 4 excellent
**Contribution:** 3 good
**Rating:** 6
**Confidence:** 3

**Summary:**

The paper proposes a new proof strategy for the convergence of Adam in the
non-convex setting. The new analysis relaxes the typical assumptions
in the following ways: 1) it assumes relaxed smoothness, where the norm of the Hessian
grows sub-quadratically with the gradient norm; 2) it does not require
bounded gradients. With a deterministic gradient oracle, the obtained rate is $O(1/\epsilon^2)$.
For the stochastic setting, the author shows a convergence rate of $O(1/\epsilon^4)$
with high probability.

Furthermore, the author proposes a variance-reduced version of Adam and proves
a rate of $O(1/\epsilon^3)$. The above rates in stochastic setting is dimension-dependent
and can be dimension-free if the Hessian norm is sub-affine on the gradient norm.

The main limitation is that although the bounded gradient assumption is removed,
the bounded noise is still needed.

**Strengths:**

1. The paper is well-written and easy to follow. Specifically, the technical
challenges and the intuition behind the new analysis are discussed comprehensively.
2. Removing the bounded gradient assumption is generally challenging. Although
there are a few works for AdaGrad that address this, existing work targeting
the removal of this assumption for Adam only converges to a neighborhood of
stationary points.
3. The paper considers the smoothness condition where the norm of the
Hessian is bounded by a sub-quadratic function of the gradient norm, which
is more general than $(L_0, L_1)$-smoothness.

**Weaknesses:**

1. Although the work relaxes the bounded stochastic gradient assumption
   commonly required in the analysis of Adam, assuming almost surely bounded
   noise is still somewhat strong and impractical. Since the proof technique,
   the contradiction argument, heavily relies on this assumption, it is
   unclear whether it can be used in more realistic settings.
2. It is difficult to assess the usefulness of the proposed variance reduction method
since the convergence result does not improve upon previous methods and there
is no experimental evidence.


Minor:

The result presented for the deterministic setting is not formal. It would be
better to have a formal theorem, at least in the appendix.

**Questions:**

1. In Theorem 4.1, the same value of $\beta_{sp}$ and $\beta$ are selected,
which differ from conventional choice of $(1 - \beta)^2 < 1 - \beta_{sp}$.
Could the author please explain the reason behind this?
2. Why do we have a dependence on $d$ in the convergence result when $\rho \geq 1$
and suddenly become dimension-free when $\rho < 1$? Also, some results for Adam
assume the infinity norm of the gradient is upper bounded by a constant [1], which
natural leads to dimension dependence. It seems that by assuming bounded 2-norm, the
results become dimension-free [2, 3]. Could the author provide further elaborate on this matter?
3. It seems that if we have access to problem-dependent parameters and the initial conditions (e.g., the gradient norm at the first iteration), SGD can also converge with the optimal rate [4]. What is the benefit of adaptive methods (if they also require tuning) compared to SGD in the more generalized smoothness setting?

References:
1. Défossez, A., Bottou, L., Bach, F., & Usunier, N. (2020). A simple convergence proof of adam and adagrad. arXiv preprint arXiv:2003.02395.
2. Chen, X., Liu, S., Sun, R., & Hong, M. (2018). On the convergence of a class of adam-type algorithms for non-convex optimization. arXiv preprint arXiv:1808.02941.
3. Guo, Z., Xu, Y., Yin, W., Jin, R., & Yang, T. (2021). A novel convergence analysis for algorithms of the adam family and beyond. arXiv preprint arXiv:2104.14840.
4. Li, H., Qian, J., Tian, Y., Rakhlin, A., & Jadbabaie, A. (2023). Convex and Non-Convex Optimization under Generalized Smoothness. arXiv preprint arXiv:2306.01264.

**Limitations:**

The limitations are addressed in the paper.

---

> ### Author Rebuttal · Authors · 2023-08-09
>
> We thank the reviewer for the insightful feedback! Below we will try to address the concerns and questions in the comments.
>
> **For the weaknesses:**
> - Regarding the bounded noise assumption, we want to first clarify that, it is not hard to generalize it to sub-Gaussian noise, as we discussed around Line 165. The contradiction argument is not really the bottleneck of further generalizing it to more heavy-tailed noise like noise with bounded variance.  The challenging part is that, when noise is heavy-tailed, there is an unignorable probability that $v_{t}$ becomes very large due to a very large noise in the gradient, which may result in a small learning rate and slow convergence. If this challenge can be overcome, we can replace the contradiction argument with something similar to the proof of convergence of VRAdam to make it work.
>
> - We also thank the reviewer for the suggestions on adding experimental results for VRAdam and formal statement in the deterministic setting and will consider adding them in the revision.
>
> **For the questions:**
>
> - 1. We choose $\beta=\beta_{sq}$ because it guarantees $\hat{v_t}\succeq\hat{m_t^2}$ (where "$\succeq$" denotes coordinate-wise inequality, see Proof of Lemma 5.2 for why it holds), which gives a uniform bound on the update $\\|x_{t+1}-x_t\\|$ and makes our analysis easier. Also, based on our preliminary experiments, Adam is not very sensitive to the momentum parameters (see Figure 1 in the PDF file attached to our global rebuttal to all reviewers).
>
> - 2. Roughly speaking, the difference is due to how we upper bound the stepsize $\frac{\eta}{\sqrt{\hat{v_t}}+\lambda}$. For $\rho<1$, the problem is relatively more smooth, and we can simply use $\frac{\eta}{\sqrt{\hat{v_t}}+\lambda}\le \frac{\eta}{\lambda}$, which to some extent makes the analysis a bit similar to SGD whose rate is dimension-free. When $\rho\ge1$, we need to also consider the $\sqrt{\hat{v_t}}$ term in the denominator and use it to bound ${\\|x_{t+1}-x_t\\|}\_{\infty}\le \eta$ as we discussed in the response to your previous question, which implies $\\|x_{t+1}-x_t\\|\_{2} \le \eta \sqrt{d}$, and gives a dimension-dependent rate. We think what we discussed here also at least partially explains why the rates in related works mentioned by the reviewer are dimension-free or dimension-dependent. Finally, we want to mention that we have some ideas on obtaining a dimension-free rate for $\rho\ge1$ with a more careful analysis and will try to make it formal in the revision.
> - 3. Since SGD already obtains an optimal rate, our results can not theoretically explain the advantage of Adam over SGD observed in practice. We will leave it as an interesting and important future work to find a reasonable condition where Adam is provably better than SGD.

---

> > ### Comment · Reviewer_rhLM · 2023-08-16
> >
> > Thanks for addressing my questions. I would like to increase my score.
> >
> > I would appreciate it if the author could elaborate more the bounded noise assumption. Regarding the response "The challenging part is that, when noise is heavy-tailed, there is an unignorable probability that  $v_t$ becomes very large due to a very large noise in the gradient, which may result in a small learning rate and slow convergence.", recent works for AdaGrad have successfully removed the need for bounded gradients and noise [1, 2]. Specifically, [2] also considers $(L_0, L_1)$-smoothness. Similar to Adam, AdaGrad may also have a large $v_t$ if there is a very large noise in the gradient. I understand the analysis for Adam would be different, but maybe some techniques from these recent works could help removing the bounded noise assumption?
> >
> >
> > **References**
> >
> > [1] Faw, Matthew, et al. "The power of adaptivity in sgd: Self-tuning step sizes with unbounded gradients and affine variance." COLT. 2022.
> >
> > [2] Wang, Bohan, et al. "Convergence of AdaGrad for Non-convex Objectives: Simple Proofs and Relaxed Assumptions." COLT. 2023.

---

> > > ### Author Response · Authors · 2023-08-19
> > > **Thanks for the reply**
> > >
> > > Thanks for the reply and further comments.
> > >
> > > We do agree that it is possible that some of the techniques from the two recent works mentioned by the reviewer could help to relax the bounded noise assumption. However, we do not think it is straightforward. One important difference between AdaGrad and Adam is that $v_t$ is non-decreasing for the former but not for the latter. In fact, [2] does rely on such property in their analysis. For example, below is a paragraph taken from their paper (page 3)
> > >
> > > *"In this paper, we propose a novel auxiliary function $\xi(t)=\frac{\nabla f(w_t)}{\sqrt{v_t}}$ for the convergence analysis of AdaGrad(-Norm), and show the error term can be bounded by $\mathbb{E}^{|\mathcal{F}_t}[\xi(t-1)-\xi(t)]$ (c.f. Lemma 4), which can be reduced by telescoping. As explained in Section 3, such an auxiliary function is rooted in the **non-increasing nature** of the adaptive learning rate $\frac{\eta}{\sqrt{v_t}}$.''*
> > >
> > > Therefore, we leave it as an interesting and potentially challenging future work.

---

> > > > ### Comment · Reviewer_rhLM · 2023-08-21
> > > >
> > > > Thanks for the explanation. My concerns have been addressed, and I have increased my score.

---

### Official Review · Reviewer_Rvyw · 2023-07-06

**Soundness:** 3 good
**Presentation:** 3 good
**Contribution:** 3 good
**Rating:** 6
**Confidence:** 4

**Summary:**

This paper studies the convergence of Adam over non-convex objectives. To begin with, this paper proposes a new non-uniform smoothness condition called $(\rho, L_0,L_1)$ smoothness condition, which generalizes $(L_0,L_1)$ smoothness condition proposed in [Zhang et al. 2019]. The authors then prove the high-probability convergence rate $\mathcal{O}(1/\sqrt{T})$ of Adam under $(\rho, L_0,L_1)$ smoothness condition and affine noise variance assumption. The authors then propose Variance Reduced Adam (VRAdam) by combining the gradient estimation of STORM into Adam, and derive the convergence rate $\mathcal{O}(1/\sqrt[3]{T^2})$ of VRAdam.

**Strengths:**

1. This paper provides the first $O(1/\sqrt{T})$ convergence rate of Adam without the bounded gradient assumption.

2. The constructed stopping time is interesting and can be of independent interest.

**Weaknesses:**

1. Although this paper provides the first $O(1/\sqrt{T})$ convergence rate of Adam without the bounded gradient assumption, the analysis is somewhat restricted because it requires a non-zero $\lambda$ and the convergence rate has a polynomial dependence over $1/\lambda$. However, in the previous analyses of Adam including [Défossez et al., 2020; Zhang et al., 2022, Wang et al., 2022], it is allowed that $\lambda=0$ or the rate has a logarithmic dependence over $1/\lambda$. I treat this as a weakness of this paper because, in practice, $\lambda$ is set to very close to $0$ (for example, $10^{-8}$ as the default value in PyTorch) and may result in very loose bound.

2.  As mentioned in this paper, the assumed noise condition is still stronger than the assumptions used for the analysis of other optimizers, for example, affine variance assumption.

**Minor Issue**:

The statement with respect to [Zhang et al., 2022; Wang et al., 2022] in line 28 is not inproper. In fact, if you pick $\beta_{sq}$ according to $\epsilon$ in [Zhang et al., 2022; Wang et al., 2022] just as this paper does, you can derive the convergence to stationary points (but with a slower rate). Therefore, a proper statement should be "but the convergence rate of [Zhang et al., 2022; Wang et al., 2022] is slower".

**References**

Defossez et al., A Simple Convergence Proof of Adam and Adagrad, 2019

Zhang et al., Adam Can Converge Without Any Modification On Update Rules, 2022

Wang et al., Provable adaptivity in Adam, 2022

**Questions:**

I will consider increasing my score according to the responses to the questions.

1. Although the studied smoothness assumption is more general, what is the motivation to study it? Specifically, according to [Zhang et al. 2019], $(L_0,L_1)$ smoothness condition seems to be enough to capture the training process of neural networks.

2. The convergence rate has a dependence on the initial gradient norm $\Vert \nabla f(x_1) \Vert $. Does this mean that Adam is sensitive to the initialization? I expect some explanations here.

---

> ### Author Rebuttal · Authors · 2023-08-09
>
> We thank the reviewer for the insightful comments. We will try to address the concerns and questions of the reviewer below.
>
> **For the weaknesses:**
> - First, regarding the dependence on $\lambda$, although it is worse than that in the papers mentioned by the reviewer, a non-zero $\lambda$ allows us to get a dimension-free rate while the existing papers mentioned by the reviewer obtain dimension-dependent rates. Given that the dimension in neural networks are usually large (especially for large language models), we think depending on $\lambda$ is at least better than depending on the dimension, because 1) $\lambda$ is a scalar hyper-parameter which is much easier to tune than the dimension; and 2) Adam is not very sensitive to $\lambda$, and a reasonably larger $\lambda$ does not make the performance worse, based on our preliminary experiments (see Figure 2 in the PDF file attached to our global rebuttal to all reviewers).
>
> - Second, regarding the noise assumption, we will leave the relaxation of it as an interesting and important future work. We also thank the reviewer for pointing out our improper statement about [Zhang et al., 2022; Wang et al., 2022] in Line 28, and will make it more precise in the revision.
>
> **For the two questions:**
> - 1. Regarding the motivation of our paper, note that Figure 1 in [Zhang et al. 2019] shows that $\log$(Hessian norm) is roughly a linear function of $\log$(gradient norm), whose slope is actually a bit larger than 1, which means Hessian norm should be a polynomial function of the gradient norm, instead of an affine function. Therefore, we believe our $(\rho,L_0,L_\rho)$ smoothness condition can better capture such relationship than the $(L_0,L_1)$ smoothness.
> - 2. Regarding the dependence on the initial gradient norm, we think it is from our analysis approach instead of the Adam algorithm , and thus does not suggest Adam is more sensitive to initialization. We want to also point out that, such dependence is not a bad thing for the following two reasons. First, for neural network training, the initial gradient norm is usually a numerical constant if you apply e.g. Kaiming initialization. Second, the initial gradient norm can be bounded by the initial sub-optimality gap $f(x_1)-f^*$ (Consider the reverse-PL inequality $\\|\\nabla f(x)\\|^2\le 2L(f(x)-f^*)$ for classical $L$-smooth functions for example, which can actually be extended to our generalized smooth function).  so the dependence on $\\|\nabla f(x_1)\\|$ can be viewed as dependence on $f(x_1)-f^*$. Most convergence results in the optimization literature do depend on $f(x_1)-f^*$.

---

> > ### Comment · Reviewer_Rvyw · 2023-08-14
> >
> > My concerns are addressed by the authors' rebuttal, and I increase my score to 6 as promised.

---

### Official Review · Reviewer_d5Nv · 2023-07-10

**Soundness:** 4 excellent
**Presentation:** 3 good
**Contribution:** 4 excellent
**Rating:** 8
**Confidence:** 2

**Summary:**

This paper studies the convergence of the Adam algorithm. Under a more general local smoothness assumption, the convergence of Adam to stationary points is proved without assuming boundedness of the loss gradient. Here the key technique is to show that the loss gradient along the trajectory is indeed bounded, using a proof by contradiction. A variance-reduced variant of Adam is proposed to achieve an accelerated gradient complexity.

**Strengths:**

The convergence of Adam is definitely an important question, and this paper seems to be a significant contribution by relaxing the bounded gradient assumption. The argument of bounding the gradients along the optimization trajectory looks neat and easy to understand. Overall the result of the paper is solid and novel, and the writing is also easy to follow.

**Weaknesses:**

It seems that the proof doesn't suggest the benefit of the momentum. It would be helpful if the authors can comment on this and provide some insights on the theoretical understanding of the momentum term.

It would also be helpful if the authors can summarize the recent results on Adam convergence in the form of a table.

**Questions:**

- Under the current local smoothness condition, how do we compare the performance of Adam and other algorithms like SGD? Does the current result suggest any advantage of Adam?

**Limitations:**

The authors have adequately addressed the limitations.

---

> ### Author Rebuttal · Authors · 2023-08-09
>
> We thank the reviewer for the positive feedback! Below we will try to address the concerns and questions of the reviewer.
>
> Although our analysis of Adam relaxes the assumptions made in previous papers, it does not provide a better theoretical understanding of the benefit of momentum or the advantage of Adam over SGD, in the nonconvex setting. We will provide some of our insights about them below.
> - 1. For the former, momentum can accelerate training for convex objective functions (e.g. Nesterov's accelerated gradient method). However, since Adam is usually applied to highly nonconvex functions like the loss for deep neural networks, we do not find it interesting or important to study Adam for convex functions. For neural network training, momentum might also help escape from sharp minima and improve generalization, which we believe is an interesting and important future direction.
>  - 2. For the latter, our intuition is that, to theoretically prove the advantage of Adam over SGD, one may need to consider different geometries other than Euclidean geometry with $\ell_2$ norm. The reason is that Adam uses coordinate-wise stepsize, which may be helpful for functions with certain geometric properties that favor coordinate-wise updates. We had some preliminary attempts like changing the $\ell_2$ norm in the definition of $(\rho,L_0,L_\rho)$ smoothness (Definition 3.2) to other norms, which unfortunately does not directly work. We will leave it as an important future work as well.
>
> We also thank the reviewer for the suggestion on summarizing the results for Adam in the literature in a table, and will do it in the revision.

---

> > ### Comment · Reviewer_d5Nv · 2023-08-14
> > **Response to the authors**
> >
> > I thank the authors for their response and their interesting discussion presented here. I don't have further questions.

---

### Official Review · Reviewer_hwio · 2023-07-25

**Soundness:** 4 excellent
**Presentation:** 3 good
**Contribution:** 3 good
**Rating:** 7
**Confidence:** 3

**Summary:**

This paper provides convergence results for Adaptive Moment Estimate (Adam) and its variance-reduced variant under generalized smooth and bouned-noise assumptions.

**Strengths:**

This paper mainly studies the convergence of Adaptive Moment Estimate (Adam) under a generalized smooth assumption. The authors drop the commonly used globally bounded gradients assumption and provide a new analysis framework, which is based on a contradiction argument, to show that gradients are bounded along the optimization trajectory. Based on this, the authors could deduce a high probability convergence bound of $\mathcal{O}(\epsilon^{-4})$ for Adam under the generalized smooth assumption. The paper also provides a variance-reduced version of Adam and improves the gradient complexity to $\mathcal{O}(\epsilon^{-3})$.

In general, the assumptions are standard in the analysis of adaptive methods including unbiased gradient estimate and bounded noise. The convergence results make sense and are highly valuable since the convergence result of Adam under unbounded gradients still leaves empty, to my best knowledge. As Adam is a widely used optimizer in deep learning field, the theoretical results (like parameter setting in Theorem 4.1 and Theorem 4.3) in this paper could help practitioners to better understand and use the algorithm. The central idea of analysis is inspiring.

**Weaknesses:**

First, the convergence result requires a dedicated tuned step-size and momentum parameters, specifically requiring prior knowledge such as generalized smooth parameter $(L_0, L_1)$ and the noise level $\sigma$. However, in most realistic situations, these parameters could be unknown or hard to obtain. Thus, the result is more valuable in the theoretical aspect but may not be so useful in the experimental aspect.

Second, the parameter setting in Theorem 4.1 contradicts the real-one default setting on deep learning packages (such as Pytorch or Tensorflow) where $\beta_1 = 0.9 $ is smaller than $\beta_2 = 0.999$. It may be more persuasive to do some experiments showing the convergence under this new parameter setting.

Third, the high probability convergence result in Theorem 6.2 does not provide an optimal rate to the probability margin $\delta$, leaving space for further improvement. In addition, since VRAdam is a new algorithm, it better requires some experiments to show its convergence under the parameter setting in Theorem 6.2 and its faster convergence rate than Adam.

**Questions:**

see the weakness part

**Limitations:**

yes

---

> ### Author Rebuttal · Authors · 2023-08-09
>
> We thank the reviewer for the positive feedback. Below we will have some discussions about the weaknesses in the comments.
>
> - First, it is true that our hyper-parameters depend on problem-dependent parameters like $\rho,L_0,L_\rho,\sigma$, and the theory does not provide many insights regarding how to choose hyper-parameters in practice in a better way than e.g. grid search. However, the main focus of this paper is to bridge the gap between theory and practice by providing convergence guarantees of Adam under more realistic assumptions than previous works.
> - Second, regarding the choice of momentum parameters $(\beta,\beta_{\text{sq}})$, we let $\beta=\beta_{\text{sq}}$ in Theorem 4.1 to make the proof easier. We have some (relatively clear) ideas on how to get rid of this requirement and will try to make it formal in the revision. In addition, Adam is not sensitive to the momentum hyper-parameters based on our preliminary experimental results (see Figure 1 in the PDF attached to our global rebuttal to all reviewers), which shows our hyper-parameter choice is as good as the default one. We will add results for more models and datasets in the revision.
> - Finally, we leave the improvement of the dependence on $\delta$ as an interesting future work, and also thank the reviewer for the suggestion about adding experiments for VRAdam, which we will consider doing in the revision.

---

> > ### Comment · Reviewer_hwio · 2023-08-16
> > **Thanks for the discussions**
> >
> > Thanks for the interesting discussions.
> >
> > Here are two remaining questions:
> >
> > 1)   Regarding the choice of momentum parameters $(\beta,\beta_{\text{sq}})$, does one need some restrictions on the relationship $\beta$ and $\beta_{sq}$ to get the convergence?  Worth noting, in [Defossez et al. 2022], they stipulated the condition $0 \le \beta_{sq} < \beta$.
> >
> > 2) There appears to be some confusion among readers regarding the intricate requirements for $\beta, \eta, G$ as outlined in Theorem 4.1. Would it be possible to present a concrete numerical parameter configuration that adheres to these conditions? Or, could the parameter choices be simplified in some specific cases?

---

> > > ### Author Response · Authors · 2023-08-19
> > > **Thanks for the reply**
> > >
> > > Below we try to answer the two remaining questions in the reply
> > > 1. We use a different analysis from existing works like [Defosses et al. 2022]. In Theorem E.1 (in the case of $0\le\rho<1$), $\beta_{sq}$ can be any constant between $0$ and $1$. In Theorem 4.1, we do require $\beta=\beta_{sq}$ for simplicity. However, as we mentioned in the rebuttal, we believe we can completely get rid of this requirement using a more careful analysis, which we will try to make formal in the revision.
> > > 2. We find it hard to further simplify the rigorous parameter choices. However, informally speaking, below are the most important requirements, which we will also add in the revision
> > >   - $G$ is a large enough constant depending on problem-dependent constants and initial gradient norm
> > >   - $\eta=O(\epsilon^2)$
> > >   - $\beta=O(\epsilon^2)$
> > >   - $T=O(\epsilon^{-4})$

---

### Author Rebuttal · Authors · 2023-08-10

In the global rebuttal, we prove some preliminary experimental results to help address the concerns of some reviewers regarding the hyper-parameters of Adam, including $\beta,\beta_{\text{sq}}, \lambda$. We train a small MLP on the Cifar10 dataset with Adam. The default parameters are $\eta=0.001, \beta=0.9, \beta_{\text{sq}}=0.999, \lambda=10^{-8}$. In the attached PDF file, we show the evolution of test errors (%) with the number of epochs.

Figure 1 shows that Adam is not sensitive to $ \beta_{\text{sq}}$, which suggests that choosing $ \beta_{\text{sq}}=\beta=0.9$ as in our Theorem 4.1 is as good as the default choice $ \beta=0.9, \beta_{\text{sq}}=0.999$. Figure 2 shows Adam is not sensitive to $\lambda$, either. So we think the dependence on $\lambda$ in our rate is not too bad, given that $\lambda=0.1$ still gives a good convergence in Figure 2.

---

### Decision · Program_Chairs · 2023-09-21

**Decision:**

Accept (spotlight)

**Comment:**

This paper shows a new analysis of Adam. Unlike standard analysis of Adam which typically requires bounded gradient and smoothness assumption, the key novelty in the analysis is a new proof of bounding gradient along the trajectory of the Adam update under a generalized smoothness condition. A variance-reduced version of Adam is also provided with a better complexity when the individual function (w.r.t. the random sample) is generalized smooth.

The reviewers unanimously supported this paper. I have read the proof sketch of the paper by myself and find the stopping time analysis novel and interesting. One minor comment is that I suggest the authors explain the order of the hyperparameters ($\eta$, $\beta$, $\lambda$, $T$) in each theorem to make readers easier to understand the complexity results. It seems that $\lambda$ has to be constant and cannot be very small (e.g., $10^{-8}$ as chosen in practice).

I recommend accepting this paper.